# Systematic Identification of External Influences in Multi-Year Micro-Seismic Recordings Using Convolutional Neural Networks

Matthias Meyer[1], Samuel Weber[2,1], Jan Beutel[1], and Lothar Thiele[1]

[1]Computer Engineering and Networks Laboratory, ETH Zurich, Zurich, Switzerland
[2]Department of Geography, University of Zurich, Zurich, Switzerland

**Correspondence:** Matthias Meyer (matthias.meyer@tik.ee.ethz.ch)

**Abstract.** Passive monitoring of ground motion can be used for geophysical process analysis and natural hazard assessment. Detecting events in micro-seismic signals can provide responsive insights into active geophysical processes. However, in the raw signals micro-seismic events are superimposed by external influences, for example anthropogenic or natural noise sources that distort analysis results. In order to be able to perform event-based, geophysical analysis with such micro-seismic data records it is imperative that negative influence factors can be systematically and efficiently identified, quantified and taken into account. Current identification methods (manual and automatic) are subject to variable quality, inconsistencies or human errors. Moreover, manual methods suffer from their inability to scale to increasing data volumes, an important property when dealing with very large data volumes as in the case of long-term monitoring.

In this work we present a systematic strategy to identify a multitude of external influence sources, characterize and quantify their impact and develop methods for automated identification in micro-seismic signals. We apply the strategy developed to a real-word, multi-sensor, multi-year micro-seismic monitoring experiment performed at the Matterhorn Hörnligrat (CH). We develop and present an approach based on convolutional neural networks for micro-seismic data to detect external influences originating in mountaineers, a major unwanted influence, with an error rate of less than 1 %, 3x lower than comparable algorithms. Moreover, we present an ensemble classifier for the same task obtaining an error rate of 0.79 % and an F1 score of 0.9383 by jointly using time-lapse image and micro-seismic data on a annotated subset of the monitoring data. Applying these classifiers to the whole experimental dataset reveals that approximately 1/4 of events detected by an event detector without such a pre-processing step are not due to seismic activity but due to anthropogenic influences and that time periods with mountaineer activity have a 9x higher event rate. Due to these findings we argue that a systematic identification of external influences using a semi-automated approach and machine learning techniques as presented in this paper is a prerequisite for the qualitative and quantitative analysis of long-term monitoring experiments.

# 1 Introduction

Passive monitoring of elastic waves, generated by the rapid release of energy within a material (Hardy, 2003) is a non-destructive analysis technique allowing a wide range of applications in material sciences (Labuz et al., 2001), engineering (Grosse, 2008) and natural hazard mitigation (Michlmayr et al., 2012) with recently increasing interest into investigations of various processes in rock slopes (Amitrano et al., 2010; Occhiena et al., 2012). Passive monitoring techniques may be broadly divided into three categories, characterized by the number of stations (single vs. array), the duration of recording (snapshot vs. monitoring) and the type of analysis (continuous vs. event-based). On the one hand, continuous methods such as the analysis of ambient seismic vibrations can provide information on internal structure of a rock slope (Burjánek et al., 2012; Gischig et al., 2015; Weber et al., 2018a). On the other hand, event-based methods such as the detection of micro-seismic events (which are focus of this study) can give immediate insight into active processes, such as local irreversible (non-elastic) deformation occurring due to the mechanical loading of rocks (Grosse and Ohtsu, 2008). However, for the reliable detection of events irrespective of the detection method the signal source of concern has to be distinguishable from noise, for example background seismicity or other source types. This discrimination is a common and major problem for analyzing micro-seismic data.

In general, event-based geoscientific investigations focus on events originating from geophysical sources such as mechanical damage, rupture or fracture in soil, rock and/or ice. These sources originate for example in thermal stresses, pressure variations or earthquakes (Amitrano et al., 2012). However, non-geophysical sources can trigger events as well: (i) anthropogenic influences such as helicopter or mountaineers (Eibl et al., 2017; van Herwijnen and Schweizer, 2011; Weber et al., 2018b) and (ii) environmental influences / disturbances, such as wind or rain (Amitrano et al., 2010). One way to account for such external influences is to manually identify their sources in the recordings (van Herwijnen and Schweizer, 2011). This procedure, however, is not feasible for autonomous monitoring because manual identification does not scale well for increasing amounts of data. Another approach is to limit to field sites far away from possible sources of uncontrolled (man-made) interference or to focus and limit analysis to decisively chosen time periods known not to be influenced by for example anthropogenic noise (Occhiena et al., 2012). In practice both the temporal limitation as well as the spatial limitation pose severe restrictions. First, research applications can benefit from close proximity to man-made infrastructure since set up and maintenance of monitoring infrastructure is facilitated (Werner-Allen et al., 2006). Second, applications in natural hazard early warning must not be restricted to special time-periods only. Moreover, they are specifically required to be usable close to inhabited areas with an increasing likelihood for human interference on the signals recorded. As a conclusion it is a requirement that external influences can be taken into account with an automated workflow, including pre-processing, cleaning and analysis of micro-seismic data.

A frequently used example of an event detection mechanism is an event detector called STA/LTA that bases on the ratio of short-term average to long-term average (Allen, 1978). Due to its simplicity, this event detector is commonly used to assess seismic activity by calculating the number of triggering events per time interval for a time period of interest (Withers et al., 1998; Amitrano et al., 2005; Senfaute et al., 2009). It is often used in the analysis of unstable slopes (Colombero et al., 2018; Levy et al., 2011) and is available integrated into many commercially available digitizers and data loggers (Geometrics,

2018). With respect to unwanted signal components, STA/LTA has also been used to detect external influence factors such as footsteps (Anchal et al., 2018) but due to its inherent simplicity, it cannot reliably discriminate geophysical seismic activity from external (unwanted) influence factors such as noise from humans, natural sources like wind, rain or hail without manually supervising and intervening with the detection process on a case by case basis. As a result the blind application of STA/LTA

will inevitably lead to the false estimation of relevant geophysical processes if significant external influences, such as wind, are present (Allen, 1978).

There exist several algorithmic approaches to mitigate the problem of external influences by increasing the selectivity of event detection. Unsupervised algorithms such as auto-correlation (Brown et al., 2008; C. Aguiar and C. Beroza, 2014; Yoon et al., 2015), but these are either computationally complex or do not perform well for low signal to noise ratios. Supervised

methods can find events in signals with low signal to noise ratio. For example template matching approaches such as cross-correlation methods (Gibbons and Ringdal, 2006) use event examples to find similar events, failing if events differ significantly in "shape" or if the transmission medium is very inhomogeneous (Weber et al., 2018b). The most recent supervised methods are based on machine learning techniques (Reynen and Audet, 2017; Olivier et al., 2018) including the use of neural networks (Kislov and Gravirov, 2017; Perol et al., 2018; Li et al., 2018; Ross et al., 2018). These learning approaches show

promising results with the drawback that large datasets containing ground truth (verified events) are required to train these automated classifiers. In earthquake research large databases of known events exist (Kong et al., 2016; Ross et al.), but in scenarios like slope instability analysis where effects are on a local scale and specific to a given field site such data are inexistent. Here, inhomogeneities are present on a very small scale and field sites differ in their specific characteristics with respect to signal attenuation and impulse response. In order to apply such automated learning methods to these scenarios obtaining a dataset of

known events is required for each new field site requiring substantial expert knowledge for a very arduous, time-consuming task. The aim of this study is to use a semi-automatic workflow to train a classifier which enables the automatic identification of unwanted external influences in real-world micro-seismic data. By this means, the geophysical phenomena of interest can be analyzed without the distortions of external influences.

To address these problems, this paper contains the following contributions. We propose a strategy to identify and deal with

unwanted external influences in multi-sensor, multi-year experiments. We compare the suitability of multiple algorithms for mountaineer detection using a combination of micro-seismic signals and time-lapse images. We propose a convolutional neural network (CNN) for source identification. We exemplify our strategy for the case of source identification on real-world micro-seismic data using monitoring data in steep, fractured bedrock permafrost. We further provide the real-world micro-seismic and image data as an annotated dataset containing data from a period of two years as well as an open source implementation

of the algorithms presented.

## 2    Concept of the classification method

In this work we present a systematic and automated approach to identify unwanted external influences in long-term, real-world micro-seismic datasets and preparing this data for subsequent analysis using a domain-specific analysis method, as illustrated

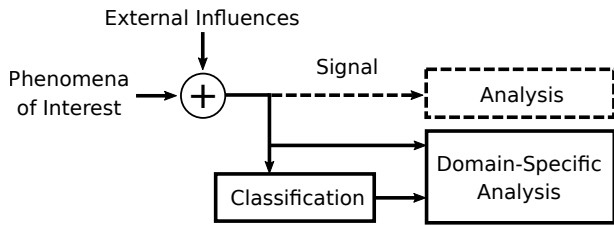

**Figure 1.** Real world measurement signals contain the phenomena of interest superimposed with external influences. If directly analyzed the results are perturbed by the external influences. In contrast to this approach (dashed lines), in this paper we suggest a systematic and automated approach to first identify a multitude of external influence sources in micro-seismic signals using a classifier. The classifier result data can then be used to quantify unwanted signal components as well as drive more extensive and powerful event detection and characterization methods leveraging combinations of both the signals as well labelled and classified noise data (solid lines).

in Fig.1. Traditionally, the signal, consisting of the phenomena of interest and superimposed external influences, is analyzed directly as described earlier. However, this analysis might suffer from distortion through the external influences. By using additional sensors like weather stations, cameras or microphones and external knowledge such as helicopter flight plans or mountain hut occupancy it is possible to semi-automatically label events originating from non-geophysical sources, such as

helicopters, footsteps or wind without the need of expert knowledge. Such "external" information sources can be used to train an algorithm that is then able to identify unwanted external influence. Using this approach multiple external influences are first classified and labeled in an automated pre-processing step with the help of state-of-the-art machine learning methods. Subsequent to this classification, the additional information can be used for domain-specific analysis for example to separate geophysical and unwanted events triggered by a simple event detector such as an STA/LTA event detector. Alternatively,

more complex approaches can be used taking into account signal content, event-detections and classifier labels of the external influences. However the specifics of such advanced domain-specific analysis methods is beyond the scope of this paper and subject to future work. A basic example of a custom domain-specific analysis method is the estimation of separate STA/LTA event rates for time periods when mountaineers are present and when they are not which we use as a case study in the evaluation section of this paper to exemplify our method.

Figure 2 illustrates the overall concept in detail. In a first step the available data sources of a case study are assessed and cataloged. Given a case study (Sect. 3) consisting of multiple sensors, one or more sensor signals are specified as primary signals (for example the micro-seismic signals, highlighted with a light green arrow in the figure) targeted by a subsequent domain-specific analysis method. Additionally, secondary data (highlighted with blue arrows) are chosen to support the classification of external influences contained in the primary signal. Conceptually these secondary data can be of different nature,

either different sensor signals, e.g. time-lapse images or weather data or auxiliary data such as local observations or helicopter flight data. All data sources are combined into a dataset. However, this resulting dataset is not yet annotated as required to perform domain-specific analysis leveraging the identified and quantified external influences.

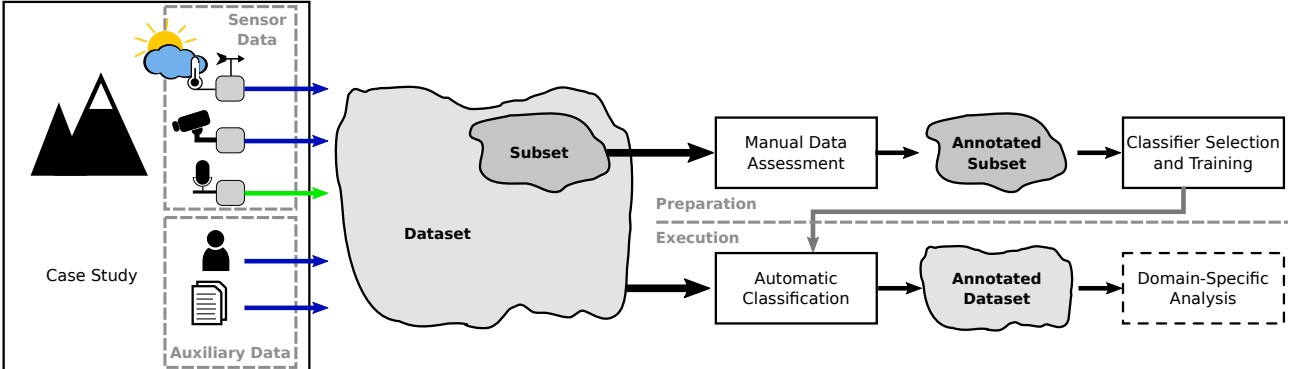

**Figure 2.** Conceptual illustration of the classification method to enable domain-specific analysis of a primary sensor signal (in our case micro-seismic signals denoted by the light green arrow) based on annotated datasets: A subset of the dataset containing both sensor and auxiliary data, is used to select and train a classifier that is subsequently applied to the whole dataset. By automatically and systematically annotating the whole dataset of the primary signal of concern, advanced methods can be applied that are able to leverage both multi sensor data as well as annotation information.

Two key challenges need to be addressed in order to establish such an annotated dataset by automatic classification: (i) suitable data types need to be selected for classification since not every data type can be used to continuously classify every external influence (for example wind sensors are not designed to capture the sound of footsteps; flight data may note be available for each time step) and (ii) a single (preferred) or at least a set of suitable, well-performing classifiers have to be found for
each type of external influence source. Once these challenges have been solved a subset of the dataset is manually annotated in order to select and train the classifier(s) in a "preparatory" phase required to be performed only once, which includes manual data assessment (Sect. 4) as well as classifier selection and training (Sect. 5). The trained classifier is then used in an automated setup to annotate the whole dataset (Sect. 6). This "execution" phase can be performed in a one-shot fashion (post-processing all data in one effort) or executed regularly, for example on a daily or weekly basis if applied to continuously retrieved real-
time monitoring data. These additional information can be used to perform a subsequent domain-specific analysis. This study concludes with an evaluation (Sect. 7) and discussion (Sect. 8) of the presented method.

## 3  Case Study

The data used in this paper originate from a multi-sensor and multi-year experiment (Weber et al., 2018b) focusing on slope stability in high-alpine permafrost rock walls and understanding the underlying processes. Specifically, the sensor data is
collected at the site of the 2003 rockfall event on the Matterhorn Hörnligrat, (Zermatt, Switzerland) at 3500 m a.s.l. where an ensemble of different sensors has monitored the rockfall scar and surrounding environment over the past ten years. Relevant for this work are data from a three-component seismometer (Lennartz LE-3Dlite MkIII), images from a remote controlled high-

resolution camera (Nikon D300, 24 mm fixed focus), rock surface temperature measurements, net radiation measurements and ambient weather conditions, specifically wind speed from a co-located local weather station (Vaisala WXT520).

The seismometer applied in the case study presented is used to assess the seismic activity by using an STA/LTA event detector, which means for our application that the seismometer is chosen as the primary sensor and STA/LTA triggering is used as the reference method. Seismic data is recorded locally using a Nanometrics Centaur digitizer and transferred daily by means of directional WLAN. The data is processed on-demand using STA/LTA triggering. The high-resolution camera's (Keller et al., 2009) field of view covers the immediate surroundings of the seismic sensor location as well as some backdrop areas further away on the mountain ridge. Figure 3 shows an overview of the field site including the location of the seismometer and an example image acquired with the camera. The standard image size is 1424x2144 pixels captured every 4 minutes. The Vaisala WXT520 weather data as well as the rock surface temperature are transmitted using a custom wireless sensor network infrastructure. A new measurement is performed on the sensors every 2 minutes and transmitted to the base station, resulting in a sampling rate of 30 samples per hour.

Significant data gaps are prevented by using solar panels, durable batteries and field-tested sensors but given the circumstances on such a demanding high-alpine field site certain outages of single sensors, for example due to power failures or during maintenance could not be prevented. Nevertheless this dataset constitutes an extensive and close-to-complete dataset.

The recordings of the case study were affected by external influences, especially mountaineers and wind. This reduced the set of possible analysis tools. Auxiliary data which helps to characterize the external influences is collected in addition to the continuous data from the sensors. In the presented case the auxiliary data is non-continuous and consist of local observations, pre-processed STA/LTA triggers from (Weber et al., 2018b), accommodation occupancy of a nearby hut and a non-exhaustive list of helicopter flight data from a duration of approximately 7 weeks provided by a local helicopter company.

In following, we use this case study to exemplify our method presented in the previous sections.

## 4 Manual Data Assessment

A ground truth is often needed for state-of-the-art classifiers (such as artificial neural networks). To establish this ground truth while reducing the amount of manual labor only a subset of the dataset is selected and used in a manual data assessment phase, which consists of data evaluation, classifier training and classifier selection as depicted in Fig. 4. Data evaluation can be subdivided into four parts: (i) characterization of external influences in the primary signal (that is the relation between primary and secondary signals), (ii) annotating the subset based on the primary and secondary signals, (iii) selecting the data types suitable for classification and (iv) performing a first statistical evaluation with the annotated dataset, which facilitates the selection of a classifier. Characterization and statistical evaluation are the only steps where domain expertise is required while it is not required for the time and labor intensive annotation process.

The classifier selection and training phase presumes the availability of a variety of classifiers for different input data types, for example the broad range of available image classifiers (Russakovsky et al., 2015). The classifiers do not perform equally well on the given task with the given subset. Therefore classifiers have to be selected based on their suitability for classification

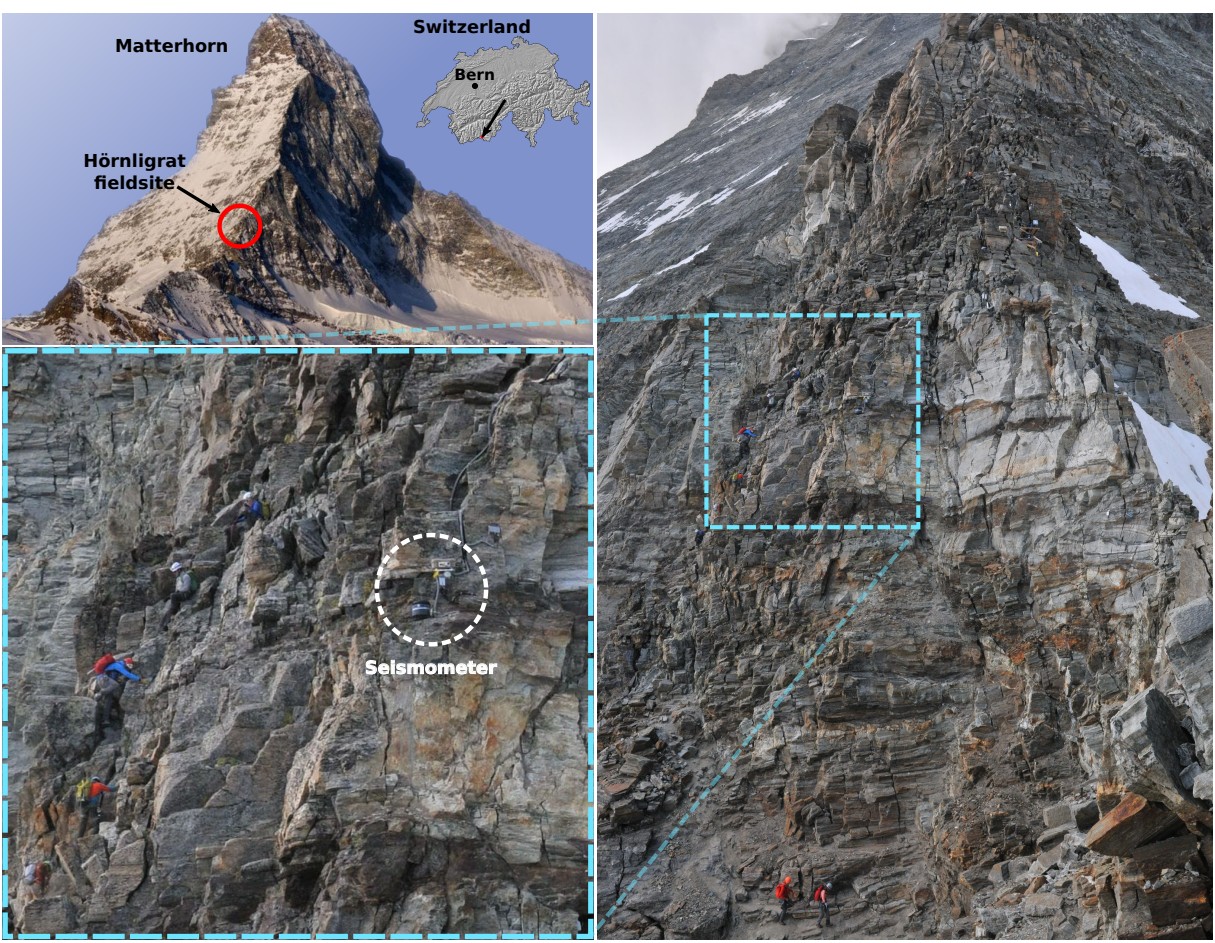

**Figure 3.** The field site is located on the Matterhorn Hörnligrat at 3500 m a.s.l. which is denoted with a red circle. The photograph on the right is taken by a remotely controlled DSLR camera on the field site at 2016-08-04T12:00:11. The seismometer of interest (white circle) is located on a rock instability which is close to a frequently used climbing route.

given the task and the data. A selection of classifiers is therefore trained and tested with the annotated subset and optimized for best performance which can for example be done by selecting the classifier with the lowest error rate on a defined test set. The classifier selection, training and optimization is repeated until a sufficiently good set of classifiers has been found. This suitability is defined by the user and can for example mean that the classifier is better than a critical error rate. These classifiers can then be used for application in the automatic classification process.

In the following, the previously explained method will be exemplified for wind and mountaineer detection using microseismic, wind and image data from a real-world experiment. The required steps of subset creation, characterization, annotation, statistical evaluation and the selection of the data type for classification are explained. Before an annotated subset can be created

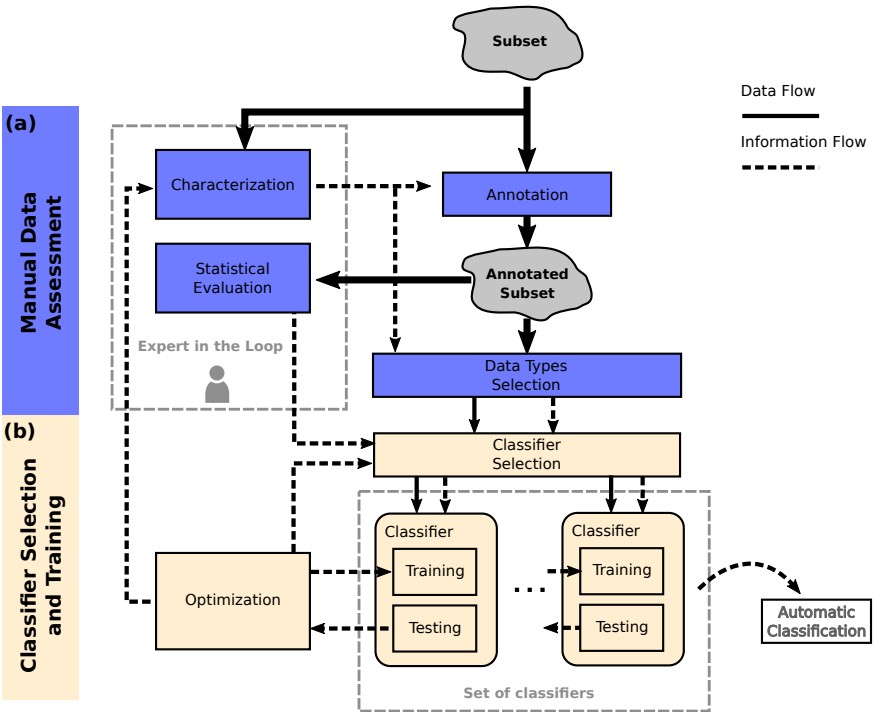

**Figure 4.** The manual preparation phase is subdivided into data evaluation (a) and classifier selection and training (b). First, the data subset is characterized and annotated. This information can be used to do a statistical evaluation and select data types which are useful for classification. Domain experts are not required for the labor intensive task of annotation. The classifiers are selected, trained and optimized in a feedback loop until the best set of classifiers is found.

the collected data must be characterized for its usefulness in the annotations process, i.e. which data type can be used to annotate which external influence.

## 4.1 Characterization

The seismometer captures elastic waves originating from different sources. In this study we will consider multiple non-geophysical sources, which are mountaineers, helicopters, wind and rockfalls. Time periods where the before mentioned sources can not be identified are considered as relevant and thus we will include them in our analysis as a fifth source, the "unknown" source. The mountaineer impact will be characterized on a long-term scale by correlating with hut accommodation occupancy (see Fig. 10) and on a short-scale by person identification on images. Helicopter examples are identified by using flight data and local observations. By analyzing spectrograms one can get an intuition what mountaineers or helicopters "look" like, which facilitates the manual annotation process. In Fig. 5 different recordings from the field site are illustrated, which have been picked by using the additional information described at the beginning of this section. For six different examples the time domain signal, its corresponding spectrogram and STA/LTA triggers are depicted. The settings for the detector are the same

for all the subsequent plots. It becomes apparent in Fig. 5 (b)-(c) that anthropogenic noise, such as mountaineers walking by or helicopters, are recorded by seismometers. Moreover, it becomes apparent that it might be feasible to assess non-geophysical sources on a larger time frame. Mountaineers for example show characteristic patterns of increasing or decreasing loudness and helicopters have distinct spectral patterns, which could be beneficial to classify these sources. Additionally, the images

captured on site show when a mountaineer is present (see Fig. 3), but due to fog, lens flares or snow on the lens the visibility can be limited. The limited visibility needs to be taken into account for when seismic data is to be annotated with the help of images. Another limiting factor is the temporal resolution of one image every 4 minutes. Mountaineers could move through the visible are in between two images.

The wind sensor can directly be used to identify the impact on the seismic sensor. Figure 6 illustrates the correlation between

tremor amplitude and wind speed. Tremor amplitude is the frequency-selective, median, absolute ground velocity and has been calculated for the frequency range of 1-60 Hz according to (Bartholomaus et al., 2015). By manually analyzing the correlation between tremor amplitude and wind speed it can be deduced that wind speeds above approximately 30 km/h have a visible influence on the tremor amplitude.

Rockfalls can best be identified by local observations since the camera captures only a small fraction of the receptive range of

the seismometers. Figure 5 (e) illustrates the seismic signature of a rockfall. The number of rockfall observations and rockfalls caught on camera are however very limited. Therefore it is most likely that we were unable to annotate all rockfall occurrences. As a consequence we will not consider a rockfall classifier in this study.

It can be seen in Fig. 5 (a) that during a period which is not strongly influenced by external influences the spectrogram shows mainly energy in the lower frequencies with occasional broadband impulses.

The red circles in the subplots in Fig. 5 indicate the timestamps of the STA/LTA events for a specific geophysical analysis (Weber et al., 2018b). By varying the threshold of the STA/LTA event trigger the number of events triggered by mountaineers can be reduced. However, since the STA/LTA event detector cannot discriminate between events from geophysical sources and events from mountaineers, changing the threshold would also influence the detection of events from geophysical sources. This fact would affect the quality of the analysis since the STA/LTA settings are determined by the geophysical application

(Colombero et al., 2018; Weber et al., 2018b).

## 4.2 Annotation

The continuous micro-seismic signals are segmented for annotation and evaluation. Figure 7 provides an overview of the three segmentation types, event-linked segments, image-linked segments and consecutive segments. Image-linked segments are extracted due the fact that a meaningful relation between seismic information and photos is only given in close temporal

proximity. Therefore images and micro-seismic data are linked in the following way: For each image a 2 minute micro-seismic segment is extracted from the continuous micro-seismic signal. The micro-seismic segment's start time is set to 1 minute before the image timestamp. Event-linked segments are extracted based on the STA/LTA triggers from (Weber et al., 2018b). For each trigger 10 seconds following the timestamp of the trigger are extracted from the micro-seismic signal. Consecutive segments are 2 minute segments sequentially extracted from the continuous micro-seismic signal.

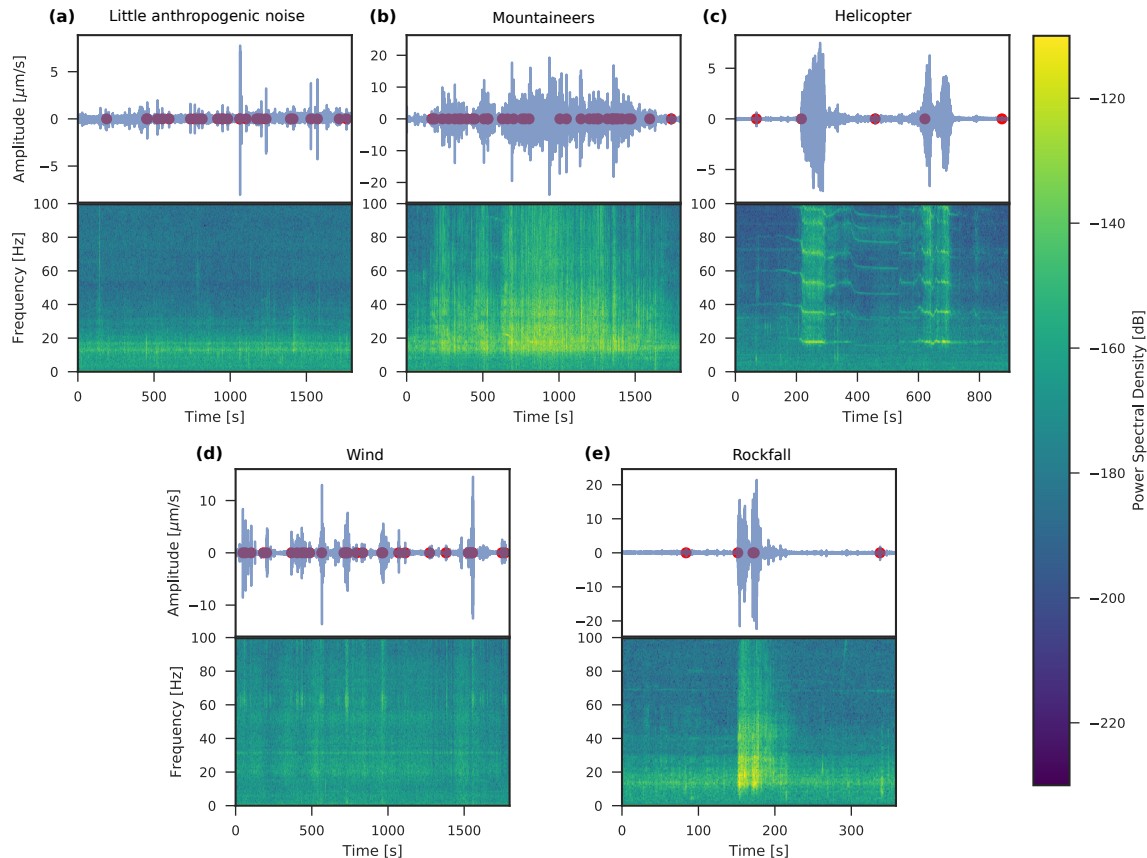

**Figure 5.** Micro-seismic signals and the impact of external influences: (a) During a period of little anthropogenic noise the seismic activity is dominant. (b) In the spectrogram the influence of mountaineers become apparent. Shown are seismic signatures of (c) a helicopter in close spatial proximity to the seismometer (d) wind influences on the signal (e) a rockfall in close proximity to the seismometer. The red dots in the signal plots indicate the timestamps of the STA/LTA triggers from (Weber et al., 2018b).

Only the image-linked segments are used during annotation, their label can however be transferred to other segmentation types by assigning the image-linked label to overlapping event-linked or consecutive segments. Image-linked and event-linked segments are used during data evaluation and classifier training. Consecutive segments are used for automatic classification on the complete dataset. Here, falsely classified segments are reduced by assigning each segment a validity range. A segment classified as mountaineer is only considered correct if the distance to the next (or previous) mountaineer is less than 5 minutes. This is based on an estimation of how long the mountaineers are typically in the audible range of the seismometer.

For mountaineer classification the required label is *mountaineer* but additional labels will be annotated which could be beneficial for classifier training and statistical analysis. These labels are *helicopters*, *rockfalls*, *wind*, *low visibility* (if the lens is partially obscured), and *lens flares*. The *wind* label applies to segments where the wind speed is higher than 30 km/h, which is the lower bound for noticeable wind impact as resulted from Sect. 4.1.

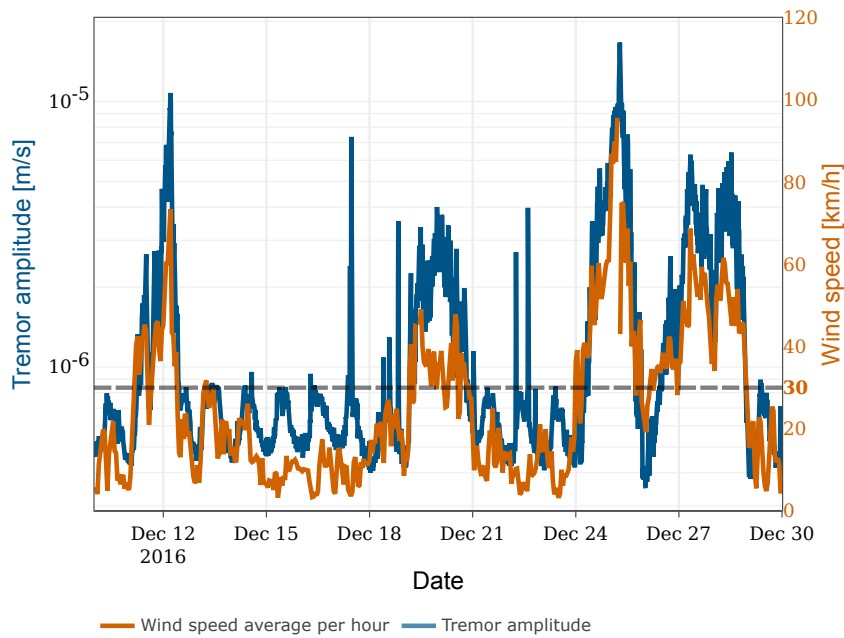

**Figure 6.** Impact of wind (light orange) on the seismic signal. The tremor amplitude (dark blue) is calculated according to (Bartholomaus et al., 2015). The effect of wind wind speed on tremor amplitude becomes apparent for wind speeds above approximately 30 km/h. Note the different scales on the y-axes.

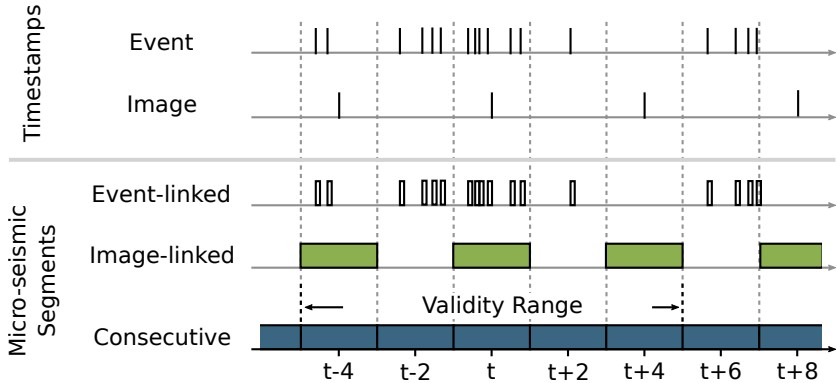

**Figure 7.** Illustration of micro-seismic segmentation. Event-linked segments are 10 second segments starting on event timestamps. Image-linked segments are two minute segments centered around an image timestamp. Consecutive segments are 2 minute segments sequentially extracted from the continuous micro-seismic signal.

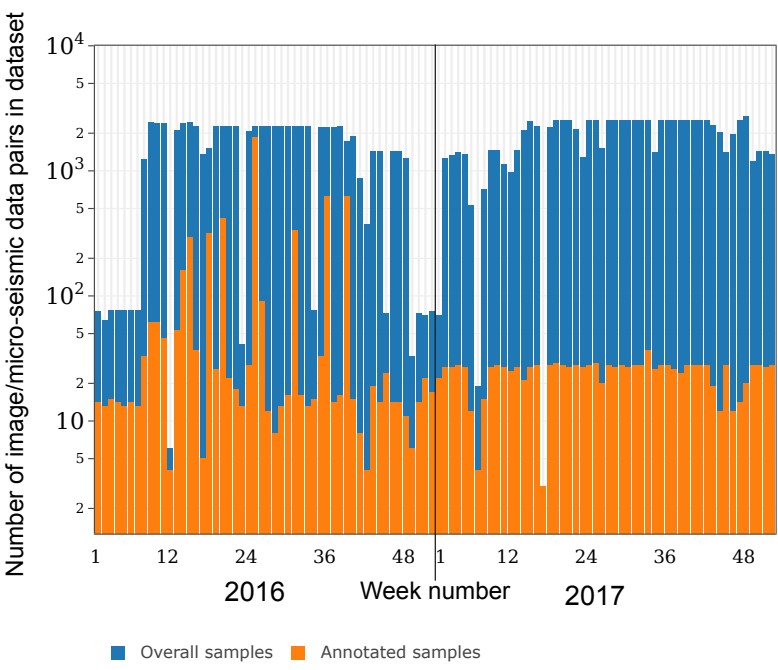

**Figure 8.** Number of image/micro-seismic data pairs in the dataset (dark blue) and in the annotated subset (light orange) displayed over the week number of the year 2016 and 2017. Note the logarithmic scale on the y-axis

Figure 8 depicts the availability of image-linked segments per week during the relevant time frame. A fraction of the data is manually labeled by the authors, which is illustrated in Fig. 8. Two sets are created, a training set containing 5579 samples from the year 2016, and a test set containing 1260 data samples from 2017. The test set has been sampled randomly to avoid any human prejudgment. For each day in 2017 four samples have been chosen randomly, which are then labeled and added

to the test set. The training set has been specifically sampled to include enough training data for each category. This means for example that more mountaineers samples come from the summer period where the climbing route is most frequently used. The number of verified rockfalls and helicopters is non-representative and although helicopters can be manually identified from spectrograms the significance of these annotations is not given due the limited ground truth from the secondary source. Therefore, for the rest of this study we will focus on mountaineers for qualitative evaluation. For statistical evaluation we

will however use the manually annotated helicopter and rockfall samples to exclude them from the analysis. The labels for all categories slightly differ for micro-seismic data and images since the type of sources which can be registered by each sensor differ. This means for instance that not every classifier uses all labels for training (for example a micro-seismic classifier cannot detect a lens flare). It also means that for the same time instance one label might apply to the image but not to the image-linked micro-seismic segment (for example mountaineers are audible but the image is partially obscured and the mountaineer is not

visible). This becomes relevant in Sect. 5.3.4 when multiple classifiers are used for ensemble classification.

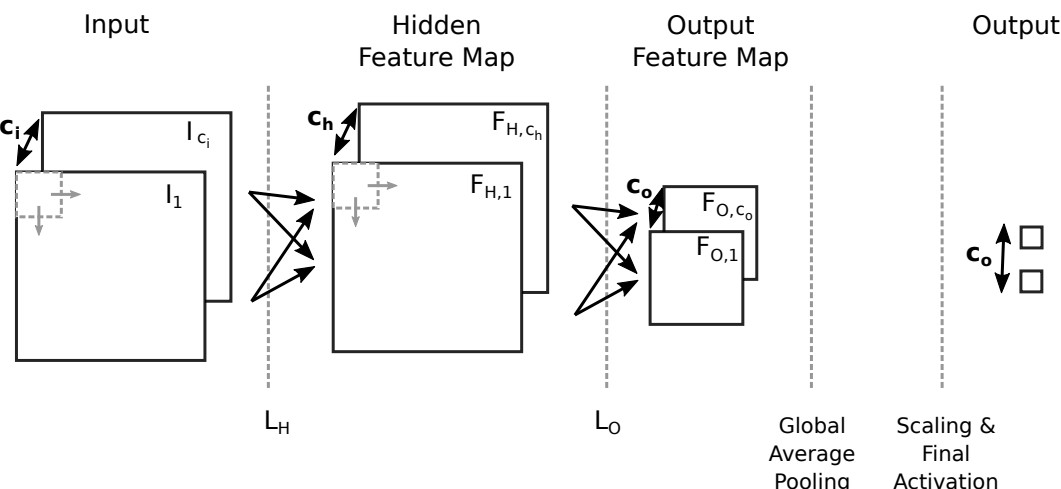

**Figure 9.** Simplified illustration of a convolutional neural network. An input signal, for example an image or spectrogram, with a given number of channels $c_i$ is processed by a convolutional layer $L_H$. The output of the layer is a feature map with $c_h$ channels. Layer $L_O$ takes the hidden feature map as input an performs a strided convolution which results in the output feature map with reduced dimensions and number of channels $c_o$. Global average pooling is performed per channel and additional scaling and a final activation are applied.

## 4.3 Data Types Selection

After the influences have been characterized the data type need to be selected which best describe each influence. The wind sensor delivers a continuous data stream and a direct measure of the external influence. In contrast, mountaineers, helicopters and rockfalls cannot directly be identified. A data type including information about these external influences needs to be selected. Local observations, accommodation occupancy and flight data can be discarded for the use as classifier input since the data cannot be continuously collected. According to Sect. 4.1 it seems possible to identify mountaineers, helicopters and rockfalls from micro-seismic data. Moreover, mountaineers can also be identified from images. As a consequence, the data types selected to perform classification are micro-seismic data, images and wind data. The micro-seismic data used are the signals from the three components of the seismometer.

## 5 Classifier Selection and Training

The following section describes the classifier pre-selection, training, testing and how the classifiers are used to annotate the whole data stream as illustrated in Fig. 4 (b). First, a brief introduction to convolutional neural networks is given. If the reader is unfamiliar with neural networks we recommend to read additional literature (Goodfellow et al., 2016).

## 5.1 Convolutional Neural Networks

Convolutional neural networks have gained a lot of interest due to their advanced feature extraction and classification capabilities. A convolutional neural network contains multiple adoptable parameters which can be updated in an iterative optimization procedure. This fact makes them generically applicable to a large range of datasets and a large range of different tasks. The convolutional neural network consists of multiple so-called convolutional layers. A convolutional layer transforms its input signal with $c_i$ channels into $c_h$ feature maps as illustrated in Fig. 9. A hidden feature map $F_{H,k}$ is calculated according to

$$F_{H,k} = g \left( \sum_{j=1}^{c_i} I_j * w_{k,j} + b_k \right)$$

where $*$ denotes the convolution operator, $g(\cdot)$ is a nonlinear function, $I_j$ is an input channel, $b_k$ is the bias related to the feature map $F_{H,k}$ and $w_{k,j}$ is the kernel related to the input image $I_j$ and feature map $F_{H,k}$. Kernel and bias are trainable parameters of each layer. This principle can be applied to subsequent convolutional layers. Additionally, a strided convolution can be used which effectively reduces the dimension of a feature map as illustrated by $L_1$ in Fig. 9. In an all convolutional neural network (Springenberg et al., 2014) the feature maps of the output convolutional layer are averaged per channel. In our case, the number of output channels is chosen to be the number of event sources to be detected. Subsequent scaling and a final (non-linear) activation function are applied. If trained correctly each output represents the probability that the input contains the respective event source. In our case, this training is performed by calculating the binary cross-entropy between the network output and the ground truth. The error is backpropagated through the neural network and the parameters are updated. The training procedure is performed for all samples in the dataset and is repeated multiple epochs.

## 5.2 Classifier Selection

Multiple classifiers are available for the previously selected data types: wind data, images and micro-seismic data.

For wind data a simple threshold classifier can be used, which indicates wind influences based on the wind speed. For simplicity the classifier labels time periods with wind speed above 30 km/h as *wind*. For images a convolutional neural network is selected to classify the presence of mountaineers in the image. The image classifier architecture is selected from the large pool of available image classifiers (Russakovsky et al., 2015). For micro-seismic data, three different classifiers will be pre-selected: (i) a footstep detector based on manually selected features (standard deviation, kurtosis and frequency band energies) using a linear support vector machine (LSVM) similar to the detector used in (Anchal et al., 2018), (ii) a seismic event classifier adopted from (Perol et al., 2018) and (iii) a non-geophysical event classifier which we call MicroseismicCNN. We reimplemented the first two algorithms based on the information from the respective papers. The third is a major contribution in this paper and has been specifically designed to identify non-geophysical sources in micro-seismic data.

The proposed convolutional neural network (CNN) to identify non-geophysical sources in micro-seismic signals uses a time-frequency signal representation as input and consists of 2D convolutional layers. Each component of the time-domain signal, sampled at 1 kHz, is first offset-compensated and then transformed with a Short-Time Fourier Transformation (STFT).

| Layer | stride | output channels |
| --- | --- | --- |
| Conv2D + BatchNorm + Linear | 1 | 32 |
| Conv2D + BatchNorm + ReLU | 2 | 32 |
| Dropout | - | 32 |
| Conv2D + BatchNorm + ReLU | 2 | 32 |
| Dropout | - | 32 |
| Conv2D + BatchNorm + ReLU | 1 | 32 |
| Conv2D + ReLU | 1 | 32 |
| Dropout | - | 32 |
| Conv2D + ReLU | 1 | 1 |
| Global Average Pooling | 1 | 1 |
| Dropout | - | 1 |
| Conv2D | 1 | 1 |
| Sigmoid Activation | - | 1 |

**Table 1.** Layout of the proposed non-geophysical event classifier, consisting of multiple layers which are executed in sequential order. The convolutional neural network consists of multiple 2D convolutional layers (Conv2D) with batch normalization (BatchNorm) and rectified linear units (ReLU). Dropout layers are used to minimize overfitting. The sequence of global average pooling layer, a scaling layer and the sigmoid activation compute one value between 0 and 1 resembling the probability of a detected mountaineer.

Subsequently, the STFT output is further processed by selecting the frequency range from 2 to 250 Hz and subdividing it into 64 linearly-spaced bands. This time-frequency representation of the three seismometer components can be interpreted as 2D signal with three channels, which is the networks input. The network consists of multiple convolutional, batch normalization and dropout layers, as depicted in Table 1. Except for the first convolutional layer, all convolutional layers are followed by batch normalization and Rectified Linear Units (ReLU) activation. Finally, a set of global average pooling layer, dropout, trainable scaling (in form of a convolutional layer with kernel size 1) and sigmoid activation reduces the features to one value representing the probability that a mountaineer is in the micro-seismic signal. In total the network has 30,243 parameters. In this architecture multiple measures have been taken to minimize overfitting: the network is all-convolutional (Springenberg et al., 2014), batch normalization (Ioffe and Szegedy, 2015) and dropout (Srivastava et al., 2014) are used and the size of the network is small compared to recent audio classification networks (Hershey et al., 2016).

### 5.3 Training and Testing

We will evaluate the micro-seismic algorithms in two scenarios in Sect. 7.1. In this section, we describe the training and test setup for the two scenarios as well as for image and ensemble classification. In the first scenario event-linked segments are classified. In the second scenario the classifiers on image-linked segments are compared. The second scenario stems from the fact that the characterization from Sect. 4.1 suggested that using a longer temporal input window could lead to a better

classification because it can capture more characteristics of a mountaineer. Training is performed with the annotated subset from Sect. 4 and a random 10 % of the training set are used as validation set, which is never used during training. For the non-geophysical and seismic event classifiers the number of epochs has been fixed to 100 and for the image classifier to 20. After each epoch the F1 score of the validation set is calculated and based on it the best performing network version is selected. The F1 score is defined as

$$\text{F1 score} = \frac{2 \cdot \text{true positive}}{2 \cdot \text{true positive} + \text{false negative} + \text{false positive}}$$

The threshold for the network's output is determined by running a parameter search with the validation set's F1 score as metric. Training was performed in batches of 32 samples with the ADAM (Kingma and Ba, 2014) optimizer and cross-entropy loss. The Keras (Chollet, 2015) framework with a TensorFlow backend (Abadi et al., 2015) was used to implement and train the network. The authors of the seismic event classifier (Perol et al., 2018) provide TensorFlow source code, but to keep the training procedure the same it was re-implemented with the Keras framework. The footstep detector is trained with scikit-learn (Pedregosa et al., 2011). Testing is performed on the test set which is independent of the training set and has not been used during training. The metrics error rate and F1 score are calculated.

It is common to do multiple iterations of training and testing to get the best performing classifier instance. We perform a preliminary parameter search to estimate the number of iterations. The estimation takes into account the number of training types (10 different classifiers need to be trained) given the limited processing capabilities. As a result of the search, we train and test 10 iterations and select the best classifier instance of each classifier type to evaluate and compare their performances in Sect. 7.

The input of the micro-seismic classifiers must be variable to be able to perform classification on event-linked segments and image-linked segments. Due to the principle of convolutional layers, the CNN architectures are independent of the input size and therefore no architectural changes have to be performed. The footstep detector's input features are averaged over time by design and are thus also time-invariant.

### 5.3.1 Event-linked Segments Experiment

Literature suggests that STA/LTA cannot distinguish geophysical sources from non-geophysical sources (Allen, 1978). Therefore the first micro-seismic experiment investigates if the presented algorithms can distinguish events induced by mountaineers from other events in the signal. The event-linked segments are used for training and evaluation. The results will be discussed in Sect. 7.1.

### 5.3.2 Image-linked Segments Experiment

In the second micro-seismic experiment the image-linked segments will be used. Each classifier is trained and evaluated on the image-linked segments. The training parameters for training the classifiers on image-linked segments are as before but additionally data augmentation is used to minimize overfitting. Data augmentation includes random circular shift and random

cropping on the time axis. Moreover, to account for the uneven distribution in the dataset, it is made sure that during training the convolutional neural networks see one example of a mountaineer every batch. The learning rate is set to 0.0001, which was determined with a preliminary parameter search. The classifiers are then evaluated on the image-linked segments.

To be able to compare the results of the classifiers trained on image-linked segments to the classifiers trained on event-linked segments (Sect. 5.3.1), the classifiers from Sect. 5.3.1 will be evaluated on the image-linked segments as well. The metrics can be calculated with the following assumption: If any of the event-linked segments which are overlapping with an image-linked segment are classified as mountaineer, the image-linked segment is considered to be classified as mountaineer as well.

The results will be discussed in Sect. 7.1.

### 5.3.3 Image Classification

Since convolutional neural networks are a predominant technique for image classification, a variety of network architectures have been developed. For this study, the MobileNet (Howard et al., 2017) architecture is used. The number of labeled images is small in comparison to the network size (approx. 3.2M parameters) and training the network on the Matterhorn images will lead to overfitting on the small dataset. To reduce overfitting a MobileNet implementation which has been pre-trained on ImageNet (Deng et al., 2009), a large-scale image dataset, will be used. Retraining is required since ImageNet has a different application focus than this study. The climbing route, containing the subject of interest, only covers a tiny fraction of the image and rescaling the image to 224x224, the input size of the MobileNet, would lead to vanishing mountaineers (compare Fig. 3). However, the image size cannot be chosen arbitrarily large since a larger input requires more memory and results in a larger runtime. To overcome this problem the image has been scaled to 448x672 pixels and although the input size differs from the pretrained version network retraining still benefits from pre-trained weights. Data augmentation is used to minimize overfitting. For data augmentation each image is slightly zoomed in and shifted in width and height. The network has been trained to detect 5 different categories. In this paper only the metrics for mountaineers are of interest for the evaluation and the metrics for the other labels are discarded in the following. However, all categories are relevant for a successful training of the mountaineer classifier. These categories consist of mountaineer, low visibility (if the lens is partially obscured), lens flare, snowy (if the seismometer is covered in snow) and bad weather (as far as it can be deduced from the image).

### 5.3.4 Ensemble

In certain cases, a sensor cannot identify a mountaineer although there is one, for example the seismometers cannot detect when the mountaineer is not moving or the camera does not capture the mountaineer if the visibility is low. The usage of multiple classifiers can be beneficial in these cases. In our case micro-seismic and image classifier will be jointly used for mountaineer prediction. Since micro-seismic labels and image labels are slightly different, as discussed in Sect. 4.2, the ground truth labels must be combined. For a given category, a sample is labeled true if any of micro-seismic or image labels are true (logical disjunction). After individual prediction by each classifier the outputs of the classifiers are combined similarly and can be compared to the ground truth.

## 5.4  Optimization

Due to potential human errors during data labeling, the training set has to be regarded as a weakly-labeled dataset. Such datasets can lead to a worse classifier performance. To overcome this issue a human-in-the-loop approach is followed where a preliminary set of classifier is trained on the training set. In the next step, each sample of the dataset is automatically classified. This procedure produces a number of true positives, false positives and true negatives. These samples are then manually relabeled and the labels for the dataset are updated based on human review. The procedure is repeated multiple times. This does however not completely avoid the possibility of falsely labeled samples in the dataset, since the algorithm might not find all human-labeled false negatives, but it increases the accuracy significantly. The impact of false labels on classifier performance will be evaluated in Sect 7.1.

## 6  Automatic Classification

In Sect. 7.1 it will be shown that the best set of classifiers are the ensemble of image classifier and MicroseismicCNN. Therefore, the trained image classifiers and MicroseismicCNN classifier are used to annotate the whole time period of collected data to quantitatively assess the impact of mountaineers. The image classifier and the MicroseismicCNN will be used to classify all the images and micro-seismic data, respectively. The consecutive segments and images are used for prediction. To avoid false positives we assume that a mountaineer requires a certain amount of time to pass by the seismometer as illustrated in Fig. 7, therefore a mountaineer annotation is only considered valid if its minimum distance to the next (or previous) mountaineer annotation is less than 5 minutes. Subsequently, events within time periods classified as mountaineer are removed and the event count per hour is calculated.

## 7  Evaluation

In the following the results of the different classifiers experiments described in Sect. 5.3 will be presented to determine the best set of classifiers. Furthermore, in Sect. 7.2 and Sect. 7.3 results of the automatic annotation process (Sect. 6) will used to evaluate the impact of external influences on the whole dataset.

### 7.1  Classifier Evaluation

The results of the classifier experiments (Table 2) show that the footstep detector is the worst at classifying mountaineers with an error rate of 0.1702 on event-linked segments and 0.0952 on image-linked segments. Both convolutional neural networks score a lower error rate on image-linked segments of 0.0096 (MicroseismicCNN) and 0.0313 (Seismic Event Classifier). For the given dataset our proposed MicroseismicCNN network outperforms the seismic event classifier, in both the event-linked segment experiment as well as the image-linked segment experiment. The MicroseismicCNN using a longer input window (trained on image-linked segments) is comparable to classification on images and outperforms the classifier trained on event-linked segments. When combining image and micro-seismic classifiers the best results can be achieved.

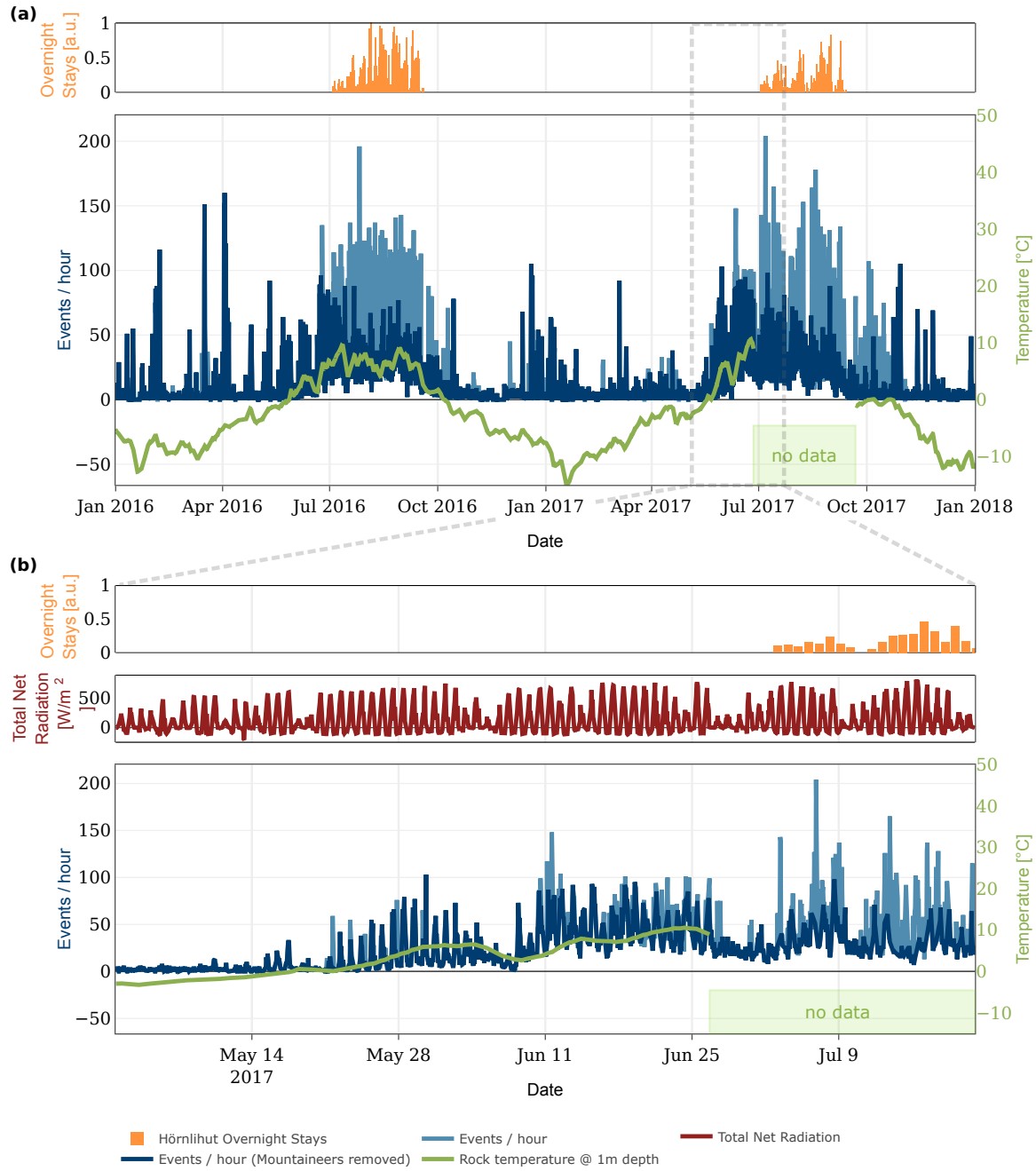

**Figure 10.** Event count, hut occupancy and rock temperature over time. (a) For the years 2016/2017 and (b) for a selected period during defreezing of the rock. The event rate from (Weber et al., 2018b) is illustrated in light blue and the rate after removal of mountaineer induced events in dark blue. The strong variations in event rate correlate with the presence of mountaineer, hut occupancy and in (b) with the total net radiation. The impact of mountaineers is significant after July 9th and event detection analysis becomes unreliable.

|  | Error Rate | F1 Score |
|---|---|---|
| **Event-linked Segments** | | |
| Footstep Detector (Events) | 0.1702 | 0.7692 |
| Seismic Event Classifier (Events) | 0.1250 | 0.8291 |
| MicroseismicCNN (Events) | **0.0641** | **0.9062** |
| **Image-linked Segments** | | |
| Footstep Detector (Events) | 0.0706 | 0.5389 |
| Seismic Event Classifier (Events) | 0.0540 | 0.6047 |
| MicroseismicCNN (Events) | 0.0309 | 0.731 |
| Footstep Detector | 0.0952 | 0.52 |
| Seismic Event Classifier | 0.0313 | 0.7383 |
| MicroseismicCNN | **0.0096** | **0.9167** |
| Image Classifier | 0.0088 | 0.9134 |
| Ensemble | **0.0079** | **0.9383** |

**Table 2.** Results of the different classifiers. The addition "(Events)" labels the classifier versions trained on event-linked segments

| False Labels (%) | 25 | 12.5 | 6.25 | 3.125 | 0 |
|---|---|---|---|---|---|
| F1 score (mean) | 0.7953 | 0.8633 | 0.8761 | 0.8835 | 0.8911 |
| Error rate (mean) | 0.0208 | 0.0149 | 0.0139 | 0.0129 | 0.0122 |

**Table 3.** Influence of falsely labeled data points on the test performance. Shown are the mean values over 10 training/test iterations.

The number of training/test iterations that were run for each classifier has been set to 10 through a preliminary parameter estimation. To validate our choice we have evaluated the influences of the number of experiments for only one classifier. The performance of the classifier is expected to depend on the number of training/test iterations (more iterations means a better chance of selecting the best classifier). However, the computing time is increasing linearly with increasing number of iterations. Hence, a reasonable trade off between the performance of the classifier and the computing time is desired to identify the ideal number of iterations. Figure 11 represents the statistical distribution of the classifier's performance for different number of training/test iterations. Each boxplot is based on ten independent sets of training/test iterations. While the box indicates the interquartile range (IQR) with the median value in orange, the whisker on the appropriate side is taken to $1.5 \times IQR$ from the quartile instead of the maximum or minimum if either type of outlier is present. Beyond the whiskers, data are considered outliers and are plotted as individual points. As can be seen in Fig. 11, the F1 score saturates at 9 iterations. Therefore our choice of 10 iterations is a reasonable choice.

In Sect. 5.4 the possibility of falsely labeled training samples has been discussed. As expected, our evaluation in Table 3 indicates that falsely labeled samples have an influence on the classification performance since the mean performances are worse for a high percentage of false labels.

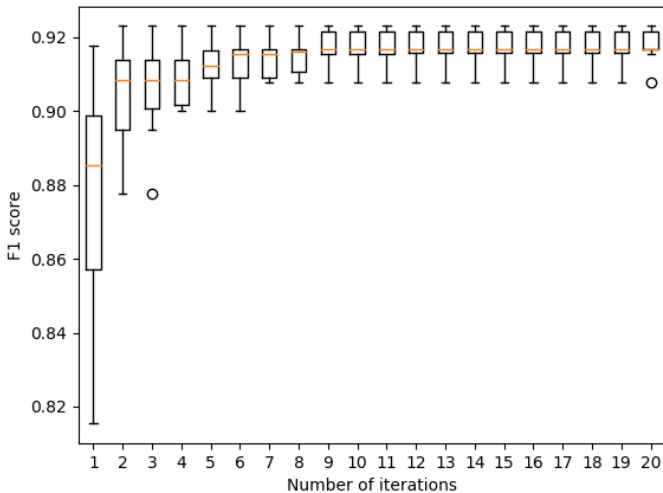

**Figure 11.** The statistical distribution of the classifier's performance for different number of training/test iterations is illustrated. Each boxplot is based on ten independent sets of training/test iterations. The F1 score saturates after 9 iterations and validates our choice of 10 iterations.

### 7.2 Statistical Evaluation

The annotated test set from Sect. 4.2 and the automatically annotated set from Sect. 6 are used for a statistical evaluation involving the impact of external influences on micro-seismic events. Only data from 2017 is chosen since wind data is not available for the whole training set due to a malfunction of the weather station in 2016. The experiment from Weber et al.
(2018b) provides STA/LTA event triggers 2017. Table 4 shows statistics for several categories, which are 3 external influences and one category where none of the 3 external influences are annotated (declared as "Unknown"). For each category, the total duration of all annotated segments is given and how many events per hour are triggered. It becomes apparent that mountaineers have the biggest impact on the event analysis. Up to 105.9 events per hour are detected on average during time periods with mountaineer activity, while during all other time periods the average ranges from 9.09 to 13.12 events per hour. This finding
supports our choice to mainly focus on mountaineers in this paper and shows that mountaineers have a strong impact on the analysis. As a consequence, a high activity detected by the event trigger does not correspond to a high seismic activity, thus relying only on this kind of event detection may lead to a false interpretation. From the automatic section in Table 4 it can be deduced that the average number of triggered events per hour for times when the signal is influenced by mountaineers is an approximately 9x increase in comparison to periods without annotated external influences. The effect of wind influences on
event rate is not as clear as the influence of mountaineers. The values in Table 4 indicate a decrease of events per hour during wind periods, which will be briefly discussed in Sect. 8.2.

As can be seen in Fig. 12, events are triggered over the course of the whole year whereas events that are annotated as coming from mountaineers occur mainly during the summer period. The main increase in event count occurs during the period when

|  |  | Unknown | Mountaineer | Wind | Helicopter |
|---|---|---|---|---|---|
| Manual | duration (hours) | 28.87 | 1.9 | 6.6 | 3.73 |
|  | mean number of events per hour | 10.6 | 95.26 | 11.21 | 13.12 |
| Automatic | duration (hours) | 6832.3 | 296.53 | 1364.2 | - |
|  | mean number of events per hour | 11.76 | 105.9 | 9.09 | - |

**Table 4.** Statistics of the manually and automatically annotated set of 2017 per annotation category. "Unknown" represents the category when none of the other categories could have been identified. Given are the total duration of annotated segments per category and the mean number of STA/LTA events per category.

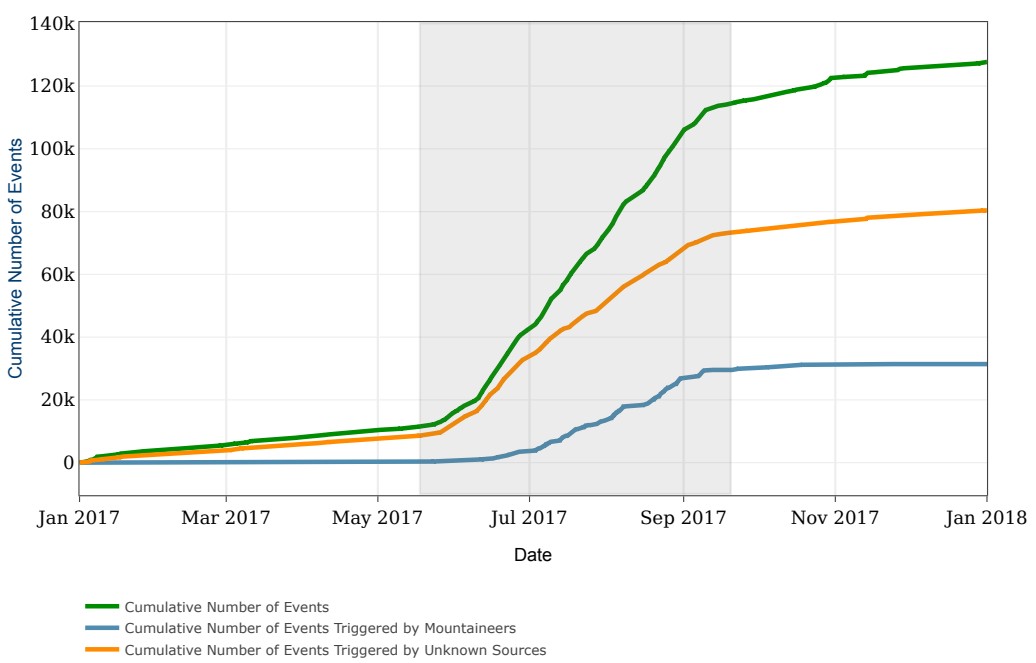

**Figure 12.** Illustration of the cumulative number of events triggered by the STA/LTA event detector for all events, for events triggered by mountaineers and for events triggered by unknown sources. The results presented in this paper were used to annotate the events. The time period during which the rock temperature in 1 m depth is above 0° C is shaded in gray.

the rock is unfrozen which unfortunately coincides with the period of mountaineer activity. Therefore it is important to account for the mountaineers. However, even if the mountaineers are not considered the event count increases significantly during the unfrozen period. The interpretation of these results will not be part of this study but they are an interesting topic for further research.

## 7.3   Automatic Annotation in a Real-World Scenario

The results of applying the ensemble classifier to the whole dataset is visualized for two time periods in Fig. 10. The figure depicts the event count per hour before and after removing periods of mountaineer activity, as well as the rock temperature, the overnight stays at the Hörnlihut and the total net radiation. From Fig. 10 (a) it becomes apparent that the mountaineer activity is mainly present during summer and autumn. An increase is also visible during increasing hut overnight stays. During winter and spring only few mountaineers are detected but some activity peaks remain. By manually review we were able to discard mountaineers as cause for most of these peaks, however further investigation is needed to explain their occurrence.

Figure 10 (b) focuses on the defreezing period. The zero-crossing of the rock temperature has a significant impact on the event count variability. A daily pattern becomes visible starting around the zero-crossing. Since few mountaineers are detected in May these can be discarded as the main influence for these patterns. The total net radiation however indicates an influences of solar radiation on the event count. Further in-depth analysis is needed but this examples shows the benefits of a domain-specific analysis, since the additional information gives an intuition of relevant processes and their interdependencies. After July 9th, the impact of mountaineers is significant and the event detection analysis becomes unreliable. Different evaluation methods are required to mitigate the influence of mountaineers during these periods.

Figure 13 depicts that mountaineer predictions and hut occupancy correlate well, which indicates that the classifiers work well. The discrepancy in the first period of each summer needs further investigation. With the annotations for the whole timespan it can be estimated that from all events detected in (Weber et al., 2018b), approximately 25% originate in time periods with mountaineer activity and should therefore not be regarded as originating from geophysical sources.

## 8   Discussion

### 8.1   Classification of Negative Examples

The previous section has shown that a certain degree of understanding of the scenario and data collected is nevertheless necessary in order to achieve a significant analysis. The effort in creating an annotated data subset, despite being time and labor consuming, is an overhead but as we show can be outweighed by the benefits of better analysis results. For data annotation two distinct approaches can be followed: Annotating the phenomena of interest (positive examples) or annotating the external influences (negative examples). Positive examples, used in (Yuan et al., 2018; Ruano et al., 2014; Kislov and Gravirov, 2017), inherently contain a limitation as this approach requires that events as well as influencing factors must be identified and identifiable in the signal of concern. This is especially hard where no ground truth information except (limited) experience by

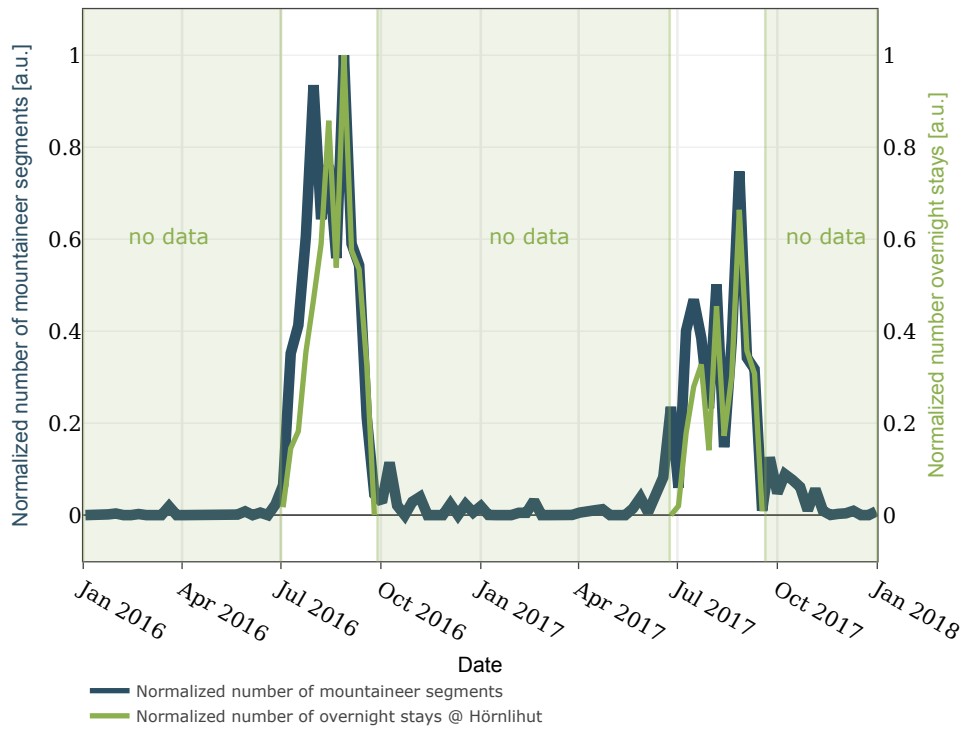

**Figure 13.** Correlation of mountaineer activity and hut occupancy. The normalized number of mountaineer segments per week and the normalized number of overnight stays at the Hörnlihut per week plotted over time.

professionals is available. Therefore, the strategy presented in this work to create an annotated dataset using negative examples is advisable to be used. It offers to perform cross checks if certain patterns can be found in different sensor/data types and in many cases the annotation process can be performed by non-experts. Also, additional sensors allow to directly quantify possible influence factors. The detour required by first classifying negative examples and then analyzing the phenomena of interest offers further benefits: In cases where the characteristics of the phenomena of interest are not known in advance (no ground truth available) and in cases where a novel analysis method is to be applied or when treating very-long-term monitoring datasets working only on the primary signal of concern is hard and error margins are likely to be large. In these cases it is important to take into account all knowledge available including possible negative examples and it is significant to automate as much as possible using automatic classification methods.

## 8.2 Multi-Sensor Classification

In Sect. 5 multiple classifiers for different sensors have been presented. The advantages of classifying micro-seismic signals are that continuous detection is possible and that no additional sensors are required. The classification accuracy of the convo-

lutional neural network and the image classifier presented are comparable. Classification of time-lapse images however has the disadvantage of a low time resolution proportional to the capture frequency, for example a maximum of 15 images per hour in our example. Continuous video recording could close this gap at the cost of requiring a more complex image classifier, the size of the data and more higher processing times, which are likely infeasible. The main advantage of images is that they can be used

as additional independent sensor to augment and verify micro-seismic recordings. First, images can be used for annotations and second they can be used in an ensemble classifier to increase the overall accuracy. The different modalities strengthen the overall meaningfulness and make the classifier more robust. Table 4 shows that during windy segments less events are triggered than in periods that cannot be categorized ("unknown" category). A possible explanation is that the micro-seismic activity is superimposed by broadband noise originating in the wind. For these time segments a variable trigger sensitivity (Walter et al.,

2008) or a different event detection algorithm can improve the analysis. Better shielding the seismometer from the wind would reduce these influences significantly but the typical approach in seismology to embed it into the ground under a substantial soil column is next to impossible to implement in steep bedrock and perennially frozen ground as found on our case-study field site. Nonetheless, Table 4 gives an intuition that our method performs well since the statistical distribution of manually and automatically annotated influences sources is similar. We therefore conclude that with our method presented it is possible to

quantify the impact of external influences on a long-term scale and across variable conditions.

## 8.3 Feature Extraction

In Sect. 4.1 the different characteristics of event sources have been discussed. The characteristic features can be used to identify and classify each source type. The convolutional neural network accomplishes the task of feature extraction and classification simultaneously by training on an extensive annotated dataset. An approach without the requirement of an annotated dataset

would be to manually identify the characteristics and then design a suitable algorithm to extract the features. For example the helicopter pattern in 5 (c) shows distinct energy bands indicating the presence of a fundamental frequency plus harmonics. These features could be traced to identify, model and and possibly localize a helicopter (Eibl et al., 2017) with the advantage of a relaxed dataset requirement. The disadvantage would be the requirement of further expertise in the broad field of digital signal processing and modeling as well as more detailed knowledge on each such phenomena class of interest. Also, it is likely

that such an approach would require extensive sensitivity analysis to be performed alongside with modeling. Moreover, if the algorithm is handcrafted by using few examples it is prone to overfitting based on these examples (see also the next subsection). This problem of overfitting exist as well for algorithm training and can be solved by using more examples, however, it is easier to annotate a given pattern (with the help of additional information) than understanding its characteristics and thus the time- and labor-consuming task of annotation can be outsourced in the case of machine learning. Fig. 5 indicates that little anthropogenic

noise (a) has less broadband background noise than wind (d) and the impulses occur in a different frequency band. However, the signal plots show a similar pattern. To identify wind from micro-seismic data manually one could utilize a frequency-selective event detector although it is not clear if this pattern and frequency range is representative for every occurrence of wind and if all non-wind events could be excluded with such a detector. Using a dedicated wind sensor for identification of wind periods

as presented in this study overcomes these issues with the drawback of an additional sensor which needs to be installed and maintained and that during failure of the additional sensor no annotation can be performed.

## 8.4 Overfitting

A big problem with machine learning methods is overfitting due to too few data examples. Instead of learning representative characteristics the algorithm memorizes the examples. In our work overfitting is an apparent issue since the reference dataset is small as described in Sect. 4.2. As explained in the previous sections multiple measures have been introduced to reduce overfitting (data augmentation, few parameters, all convolutional neural network, dropout). The test set has been specifically selected to be from a different year to exclude that severe overfitting affects the classifier performance. The test set includes examples from all seasons, day and night time and is thus assumed representative for upcoming, never-seen-before data. However, overfitting might still exist in the sense that the classifier is optimized for one specific seismometer. Generalization to multiple seismometers still needs to be proven since we did not test the same classifier for multiple seismometers, which might differ in their specific location, type or frequency response. This will be an important study for the future since it will reduce the dataset collection and training time significantly if a new seismometer is deployed.

## 8.5 Outlook

This work has only focused on identifying external influences, what we have shown to be a prerequisite for micro-seismic analysis. Future work lies in finding and applying specific analytic methods, especially finding good parameter sets and algorithms for each context. Additionally, the classifier could be extended to include helicopters as well as geophysical sources such as rockfalls. A disadvantage of the present method is the requirement of a labeled dataset. Semi-supervised or unsupervised methods (Kuyuk et al., 2011) as well as one- or few-shot classification methods (Fei-Fei et al., 2006) could provide an alternative to the presented training concept without the requirement of a large annotated dataset.

## 9 Conclusions

In this paper we have presented a strategy to evaluate the impact of external influences on a micro-seismic measurement by categorizing the data with the help of additional sensors and information. With this knowledge a method to classify mountaineers has been presented. We have shown how additional sensors can be beneficial to isolate the information of interest from unwanted external influences and provide a ground truth in a long-term monitoring setup. Moreover, we have presented a mountaineer detector, implemented with a convolutional neural network, which scores an error rate of only 0.96 % (F1 score: 0.9167) on micro-seismic signals and a mountaineer detector ensemble which scores an error rate of 0.79 % (F1 score: 0.9383) on images and micro-seismic data. The classifiers outperform comparable algorithms. Their application to a real-word, multi-sensor, multi-year micro-seismic monitoring experiment showed that time periods with mountaineer activity have a approximately 9x higher event rate and that approximately 25% of all detected events are due to mountaineer interference. Finally, the findings of

this paper show that an extensive, systematic identification of external influences is required for a quantitative and qualitative analysis on long-term monitoring experiments.

*Code and data availability.* The dataset is available under the DOI 10.5281/zenodo.1320835 and the accompanying code under the DOI 10.5281/zenodo.1321176.

5 *Author contributions.* Matthias Meyer, Samuel Weber, Jan Beutel and Lothar Thiele developed the concept. Matthias Meyer and Samuel Weber developed the code and maintained field site and data together with Jan Beutel. Matthias Meyer prepared and performed the experiments and evaluated the results with Samuel Weber. Matthias Meyer prepared the manuscript as well as the visualizations with contributions from all co-authors.

*Competing interests.* The authors declare that they have no conflict of interest.

10 *Acknowledgements.* The work presented in this paper is part of the X-Sense 2 project. It was financed by http://www.nano-tera.ch (ref. no. 530659). We would like to thank Tonio Gsell and the rest of the PermaSense team for technical support. We acknowledgement Kurt Lauber for providing us with hut occupancy data and the Air Zermatt helicopter company for providing us with helicopter flight data. We thank Lukas Cavigelli for insightful discussions. Reviews from Marine Denolle and two anonymous referees provided valuable comments that helped to improve the paper substantially. Finally, we thank the handling Editors Tom Coulthard and Jens Turowski for constructive feedback and 15 suggestions.

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
