# Peer review of "Systematic Identification of External Influences in Multi-Year Micro-Seismic Recordings Using Convolutional Neural Networks"

_Earth Surface Dynamics, 2018_

## Short Comment (SC1) · 13 Aug 2018

For some figures an interactive version exists which allows to zoom in on specific details. These figures can be generated with the provided assets. For convenience I have uploaded them to my personal website.

http://people.ee.ethz.ch/~matthmey/post/influence_identification/

---

## Referee Comment (RC1) · Anonymous Referee #1 · 24 Aug 2018

This paper addresses the issue of accurate attribution of seismic events to the correct source in long-term/large (micro-)seismic datasets. This paper has the potential to form a helpful methodological contribution to the geomorphic literature, and the overall result is promising. However, I do not believe the paper is ready for publication in its current format. Whilst there is some interesting information presented here, the focus, clarity and structure of the paper require further work.

General points

The language is often vague, with loose use of specific terminology. For example, in the abstract, the authors mention that '...Successful analysis depends strongly on the

capability to cope with such external influences'. What do they mean by 'successful analysis' and 'coping' with these influences? Similarly, the authors mention 'correct slope characterisation' in the next sentence. What does this mean? It suggests consideration of the structural/strength/geometric properties and/or damage condition of the slopes. It is not clear which the authors are addressing, and why. Linked to this, Fig. 5 suggests that the focus of the paper is on rockfalls, which again is different to 'slope characterisation'. In short, what is the geomorphic nature of the seismic activity the authors are considering? The final sentence in the abstract is also rather obvious and can be made without the detailed assessment presented in the paper. Indeed, this type of source characterisation is commonly done (and done well) by geomorphologists (see e.g. the work of Adam Young on coastal microseismic monitoring). The most interesting part here is the ability to distinguish between sources of microseismic activity in large/long-term monitoring datasets, and this needs to be more clearly presented.

The Introduction repeats the same points multiple times in subtly different ways – this section could be condensed considerably. Links to geomorphic processes are implicit at best, and largely absent. For example, what exactly are you trying to monitor? Rockfall occurrence? Ground cracking and associated micro-seismic signal? This isn't clear. There is also a stark lack of reference to appropriate literature (e.g. page 2, lines 10 – 16). The aim of the paper is not clear and the authors present instead a bullet-point list of study conclusions. What is the focus here and what is novel?

Section 2 again repeats much of what we have already been told in the introduction. The methods section is again repetitive, justifying the need for, and broad benefits of, the approach, rather than stating concisely how it works. Much of the information here is not clear. For example, Page 6, Lines 16 -21 - there is no specific detail about how tasks are undertaken and how 'a good set of classifiers' is objectively specified.

Much of the methods section lacks detail and feels very descriptive and subjective; many of the choices made are not fully/objectively demonstrated. For example, Fig. 6

does not clearly demonstrate the wind speed threshold required for a 'visible influence' on tremor amplitude. Important definitions do not appear in logical places (e.g. tremor amplitude is defined after it has been used in the text). The key aspect of the event trigger threshold by STA/LTA is not appropriately addressed; I would like to see more critique of the application of the method in this setting. Is it too sensitive and/or appropriate given the plots in Fig. 5? How is the accuracy of event attribution assessed, other than by ruling out mountaineers etc. and process of elimination (page 9, lines 8 – 10 suggests this is the case)? Sections 2.4.2 –3.4 contain some ostensibly important methodological steps, but again much of these sections feels descriptive, lacks an appropriate justification and a logical structure to follow the workflow and the choices made. The level of assumed knowledge about neural network is also rather high. I am not convinced by the 'statistical analysis' presented in Table 1 – this seems rather weak and limited in terms of the depth of data analysis.

The results section draws out the key argument that the authors wish to make, but I would like to see more assessment of the data presented in Fig. 5, even at the basic level, including the duration and frequency range/spectral density of different seismic sources. Can this information be used in a simpler manner to draw the same conclusions? How sensitive are the patterns shown by the graphs to the colour scale of the spectral density information?

The discussion section is underdeveloped, lacks grounding and critique in the context of related literature and does not address the geomorphic significance of the approach addressed. How does the constrained uncertainty of the approach considered compare to other sources of uncertainty, such as seismometer tilt (indeed, which component of the seismometer is being used, and why? – again see the work of Adam Young) and rock slope resonance and site effects (see e.g. Burjanek et al, 2012, 2017 GJI)? Some of the claims made about trade-off between time and accuracy feel poorly considered and require a more robust demonstration. There is also no discussion of the representivity of the case studies provided and how changes in the nature of the rock

mass may affect the accuracy/source attribution of the seismic readings through time (e.g. resonance effects on duration and frequency as a rock mass degrades).

Specific comments (not exhaustive)

There are many uses of e.g. in the manuscript – remove these and replace with 'such as' or 'for example' as appropriate.

Fig. 5 - what are the red/purple circles? Are these triggered microseismic events? This isn't clear in the figure or the caption.

Brackets for citations are not always used correctly. Please check and amend.

Line 19 – check terminology. Rockfalls are a type of landslides (see e.g. Varnes, 1978, and subsequent iterations of this work).

Line 1 – what is the difference between acoustic emission and micro-seismic emission? Clarify.

Line 3 – . . .HAS been demonstrated. . .

Line 5 – micro-seismic RECORDS?

Line 6 – biased assessments of what?

Lines 3 – 4 and 5 – 9 – very repetitive.

Line 9 – expand on scaling issues.

Lines 18 – 19 – Do you mean the accurate attribution of seismic events?

Line 21 – what is the significance of footsteps?

Line 22 – its (not it's)

Line 16 – F1 is not defined at this point. Indeed, much of the terminology in this section (e.g. ensemble classifier) is not clearly defined.

Line 28 - ...has monitored...? Tense is not correct.

Line 19 – triaxial or three-axis?

Lines 6/7 – clarify 'sampling rate' – how was the sampling done? Or are you referring to the data transmission interval?

Line 11 – The microseismic records considered in the case study were affected...?

Line 8 – expand on 'scaling issues' – this is unclear.

Line 24 – is 'sounds' the correct word here?

Lines 26 – 29 – the distinction between acoustic events and seismic events is confusing, seems a little arbitrary and lacks reference to the literature; some of the terms do not follow some conventions in e.g. laboratory monitoring of acoustic emissions; this is important for a contribution to the geophysical literature. These definitions and distinctions also come too late in the manuscript, since these terms are used earlier.

Line 32 – sentence beginning 'Additionally...' is not clear.

Line 9 – Figure 5(e) does not show an example of a rockfall. It shows an example of the seismic signature of a rockfall event.

Line 5 – do not use comma splices (re: therefore)

Table 1 – Reword the caption. It is not the case that none of the other categories 'apply'. Rather, it is where you have not been able to classify the signal as one of the three categories discussed.

Table 2 – this needs a lot more detail – what is this showing?

---

## Referee Comment (RC2) · Anonymous Referee #2 · 13 Sep 2018

This paper proposed a new method for identifying external influences such as winds or mountaineers in micro-seismic recordings. Because the external influences may cause bias interpretations, its identification is very important for understanding micro-seismic recordings. In addition, the method may help to interpret the external influences which are keys to improve our understanding of rock-slope failure processes. The similar idea using machine learning is already applied to seismic wave discrimination such as Li et al., 2018. This study is interesting and suitable for the publication after moderate revisions. I suggest the authors revise this manuscript and pay attention to the following list as general suggestions: 1. Acknowledge previous studies on this topic or related topics and make sure the readers understand your contribution; 2. Introduce more

about methods especially for Convolution Neural Network since readers may not be familiar with this method at all; 3. Discuss more future works such as how to automatically learn signal pattern in the external influences to improve the classification and interpretation of rock-slope failure processes; Here are more specific comments: Page 9, line 1-4: I am confused about this part. Does this part mean that the dataset may be mislabeled due to fog, lens flares or other reasons? Have this data been included in the training dataset? Similar problem for the rockfalls in line 8-10?

Page 11, line 12-15, the dataset including training dataset and test dataset seems to be small and may have serious overfitting problem. The authors need to address this issue during the discussion part and prove the trained model can handle it well.

Page 14, line 12-14 The results with ten iterations are presented in this paper, but it will be better to show how the results change for a different number of iterations (such as 1, 5, 10, 20 iterations).

Page 15, Line 10: The learning rate is very small, which may make the code very slow. Is there any specific reason to set this small value?

Page 16, line 10-16: Since it needs to manually relabel for the dataset in some cases, it will be worth to discuss how the potential human errors during data labeling will influence the classifier performance.

Page 20 line 14-22: The method is trained based on negative examples. But in most conditions, we should pay more attention to the phenomena of interest. In the discussion part, the author should discuss how this method can improve our interpretation of the processes of interest.

Perol, Thibaut, Michaël Gharbi, and Marine Denolle. "Convolutional neural network for earthquake detection and location." Science Advances 4.2 (2018): e1700578.

Ross, Zachary E., Men‐Andrin Meier, and Egill Hauksson. "P‐wave arrival picking and first‐motion polarity determination with deep learning." Journal of Geophysical Research: Solid Earth (2018).

Olivier, Gerrit, Julien Chaput, and Brian Borchers. "Using supervised machine learning to improve active source signal retrieval." Seismological Research Letters 89.3 (2018): 1023-1029.

Li, Zefeng, et al. "Machine Learning Seismic Wave Discrimination: Application to Earthquake Early Warning." Geophysical Research Letters (2018).

---

## Editor Comment (EC1) · JM Turowski (Editor) · 28 Sep 2018

Dear authors,

I agree with the reviewers that the papers presents a potentially very useful contribution, however, currently there are weaknesses and the presentation (language and structure). The comments of the reviewers largely speak for themselves; although they assess the paper from different angles, the common feature of their assessment is that there is a lack of clarity. Reviewer #1 focusses more on the language and structure problems. In addition, he asks to evaluate your work in a broader context in the discussion/conclusion. Similarly, reviewer #2 asks for a more detailed description of the

method and a better acknowledgment of previous work.

I largely agree with that. When revising the paper, I ask you specifically to think from the perspecitve of a potential user. Is all the information available to reproduce the analysis? Can the reasoning be followed easily? Can the necessary information be clearly accessed in the manuscript? Is the approach described separately from specific features of the case study? Is the evaluation of the method objective?

In addition, I want to highlight a few specific stylistic points, which may help you to revise the paper and achieve the aims outlined above. I will do this quoting specific example (page.line).

2.11 '...very unattractive overall solution...': please avoid subjective judgements such at this. It is better to list the pros and cons, and then explain your priorities and your reasoning.

5.3 'Care is taken...'; 6.12 'But the meticulous care does pay off.'; 7.8 'Care has been taken to prevent significant data gaps...': Such statements are not helpful to the reader. The phrasing is meant to convey some particularly high standards of scientific rigour. However, it is unclear what you have actually done, and thus your 'care' is not reproducible. It would be better if you describe your actual measures (e.g., for preventing significant data gaps) and then describe how well they worked, and if they failed, why they failed. Again, here it is important to keep the reproducability of the work in mind.

8.6 'It becomes apparent in Fig. 5 (b)-(c) that anthropogenic noise, such as mountaineers walking by or helicopters, can have a strong influence on seismic recordings.': This sentence is an example of how results are mixed into the method description. There are multiple other instances. I ask you to separate this and present the methodology in the methods section and the results in the results section.

18.2 'The results of the classifier experiments from Sect. 3.2 are listed in Table 3.': Such sentences contain little information. It would be better to state the main result or

feature (that is important in the current context) and then cite the table in parentheses.

Looking forward to seeing your revised paper,

best wishes, Jens Turowski

---

## Author Comment (AC1) · 16 Nov 2018

article [utf8]inputenc

*RC: Referee Comment*
AC: Author Comment

[Figure]

**1   Response to RC1**

AC: We would like to thank the anonymous reviewer for the extensive review and the valuable feedback. We will incorporate the feedback and address all comments in the following. For a preliminary revised manuscript highlighting all the modifications please refer to the response to the editor.

*RC: This paper addresses the issue of accurate attribution of seismic events to the correct source in long-term/large (micro-)seismic datasets. This paper has the potential to form a helpful methodological contribution to the geomorphic literature, and the overall result is promising. However, I do not believe the paper is ready for publication in its current format. Whilst there is some interesting information presented here, the focus, clarity and structure of the paper require further work.*

AC: We restructured the manuscript to make it more precise in terms of terminology and the geoscientific field we consider. Moreover, we have condensed the information to make it more accessible.

*RC: General points The language is often vague, with loose use of specific terminology. For example, in the abstract, the authors mention that 'Successful analysis depends strongly on the capability to cope with such external influences'. What do they mean by 'successful analysis' and 'coping' with these influences?*

AC: We acknowledge the loose use of specific terminology in the initial submission. In the revised manuscript we use a more precise terminology and avoided misleading formulations, such as 'successful analysis' mentioned in your comment.

*RC: Similarly, the authors mention 'correct slope characterisation' in the next sentence. What does this mean? It suggests consideration of the structural/strength/geometric properties and/or damage condition of the slopes. It is not clear which the authors are addressing, and why. Linked to this, Fig. 5 suggests that the focus of the paper is on rockfalls, which again is different to 'slope characterisation'. In short, what is the geomorphic nature of the seismic activity the authors are considering?*
*RC: Links to geomorphic processes are implicit at best, and largely absent. For example, what exactly are you trying to monitor? Rockfall occurrence? Ground cracking and associated micro-seismic signal? This isn't clear. There is also a stark lack of reference to appropriate literature (e.g. page 2, lines 10 – 16).*

AC: In contrast to the submitted manuscript, we have precisely defined the application context in the revised manuscript. The context is geophysical analysis on micro-seismic signals in general and event-based analysis in particular. The specific analysis performed using event-based methods depends on the use case and is not the focus of our study. For example the case study we use to demonstrate our method focuses on rock fracturing. We have also added additional literature to define the different geophysical applications and state more precisely which application we focus on: (Hardy, 2003), (Michlmayr et al., 2012), (Gischig et al., 2015), (Burjánek et al., 2012), (Weber et al., 2018).

*RC: The final sentence in the abstract is also rather obvious and can be made without the detailed assessment presented in the paper. Indeed, this type of source characterization is commonly done (and done well) by geomorphologists (see e.g. the work of Adam Young on coastal microseismic monitoring). The most interesting part here is the ability to distinguish between sources of microseismic activity in large/long-term monitoring datasets, and this needs to be more clearly presented.*

AC: You are correct that the final sentence "Due to these findings we argue that a systematic identification of external influences, like presented in this paper, is a prerequisite for a qualitative analysis." is rather obvious. Indeed, we want to show in this paper how the source characterization can be done systematically for a large and long-term monitoring experiment. Consequently, we have updated the statement to be more precise: "Due to these findings we argue that a systematic identification of external influences using a semi-automated approach and machine learning techniques as presented in this paper is a prerequisite for the qualitative and quantitative analysis of long-term monitoring experiments."

*RC: The Introduction repeats the same points multiple times in subtly different ways – this section could be condensed considerably.*
*RC: Section 2 again repeats much of what we have already been told in the introduction.*
*RC: Sections 2.4.2 –3.4 contain some ostensibly important methodological steps, but again much of these sections feels descriptive, lacks an appropriate justification and a logical structure to follow the workflow and the choices made.*
*RC: The aim of the paper is not clear and the authors present instead a bullet-point list*

*of study conclusions. What is the focus here and what is novel?*

AC: The introduction has been condensed significantly. Moreover, due to the reorganization of the first sections the reading flow has been improved. Whereas the initial manuscript had a potentially confusing structure, the updated manuscript follows a clear structure of

- Introduction
- Concept of the classification method
- Case Study
- Manual Data Assessment
- Classifier Selection and Training
- Automatic Classification
- Evaluation
- Discussion
- Conclusions

The aims of the paper are now highlighted at the end of the introduction with precise statements about the contributions and novelty.

*RC: The methods section is again repetitive, justifying the need for, and broad benefits of, the approach, rather than stating concisely how it works. Much of the information*

*here is not clear. For example, Page 6, Lines 16 -21 - there is no specific detail about how tasks are undertaken and how 'a good set of classifiers' is objectively specified. Much of the methods section lacks detail and feels very descriptive and subjective; many of the choices made are not fully/objectively demonstrated.*

AC: We have condensed the methods section (now called "Manual Data Assessment") significantly and concisely described the steps required for manual data assessment. A detailed description to objectively demonstrate our choices is given in each subsection. Additionally, to avoid the "descriptive feel" we have added specific examples to the classifier selection description (see p. 6 l. 24 - 31). In this way it remains clear which steps are to be taken on a high, methodology level while having short specific details on e.g. 'how a good set of classifiers' is specified (which is later defined in detail in the respective subsection).

*RC: For example, Fig. 6 does not clearly demonstrate the wind speed threshold required for a 'visible influence' on tremor amplitude. Important definitions do not appear in logical places (e.g. tremor amplitude is defined after it has been used in the text).*

AC: For better understanding we have indicated the wind speed threshold in the respective figure. Moreover, we have rephrased the section about the tremor amplitude such that the explanation is directly available to the reader (see Page 9, Lines 1 - 5).

*RC: The key aspect of the event trigger threshold by STA/LTA is not appropriately addressed; I would like to see more critique of the application of the method in this setting. Is it too sensitive and/or appropriate given the plots in Fig. 5?*

AC: We discussed the STA/LTA characteristics throughout the submitted manuscript. In the revised manuscript we added an additional paragraph to discuss the STA/LTA settings in the context of Fig.5, including the effect of the threshold (see Page 9, Lines 12 - 17). We hope that thereby the application of the method in this setting are described in greater detail.

*RC: How is the accuracy of event attribution assessed, other than by ruling out mountaineers etc. and process of elimination (page 9, lines 8 – 10 suggests this is the case)?*

AC: We assume that the term event in this question relates to geophysical events for example the rockfalls discussed on page 9, lines 8 – 10 in the initial submission. In this case the accuracy of the attribution is related to the accuracy of our sources, which are incidents reported and logged by local observers, for example during maintenance of the monitoring setup. In addition, some rockfalls can also be seen by analyzing image sequences. We relate the characteristics of the micro-seismic signal to the timestamps of a rockfall report containing beginning and end timestamp. We do not use additional information/knowledge about a characteristic signal pattern which makes a rockfall identifiable only with the micro-seismic signal. Since we take only verified events into account the accuracy of these events is rather high but it also means that we probably missed to annotate rockfalls occurrences during the two years. As a result, we did not use our classifier to automatically annotate rockfall occurrences since the dataset is not accurate enough to train a rockfall classifier. Additionally, to evaluate the accuracy

of event attribution even more we introduce a new evaluation which investigates how false labels affect the classifier performance. This evaluation is presented and discussed in the section "Classifier Evaluation".

*RC: The level of assumed knowledge about neural network is also rather high.*

AC: We have added a new subsection "Convolutional Neural Networks" (starting Page 9, Line 7) to explain the concept of convolutional neural networks and we recommend additional literature for the interested reader

*RC: I am not convinced by the 'statistical analysis' presented in Table 1 – this seems rather weak and limited in terms of the depth of data analysis.*

AC: We have restructured the statistical evaluation and improved the analysis. First, methods and evaluation are strictly separated into their individual section. Second, the "Statistical Evaluation" section now comprises all results from the initial submission and new results we have added as a consequence of your comment. The results presented and discussed in depth are (i) statistics for the manually annotated test set, (ii) statistics for the automatically annotated set for the year 2017, (iii) a plot which illustrates the distribution of STA/LTA events over time.

*RC: The results section draws out the key argument that the authors wish to make, but I would like to see more assessment of the data presented in Fig. 5, even at the basic level, including the duration and frequency range/spectral density of different seismic sources. Can this information be used in a simpler manner to draw the same conclusions?*

AC: In the new section "Feature Extraction" we have addressed the before mentioned comment. We make an assessment of the different source characteristics and how these can be used to classify/distinguish event sources. Moreover, we discuss the pro/cons of classifying based on manually extracted features versus classifying on learned features.

*RC: How sensitive are the patterns shown by the graphs to the colour scale of the spectral density information?*

AC: The color scale is only a visualization guideline and has been set to the same range for all subplots to maintain comparability between the subplots. The visibility of the patterns would change with a different scale but please note that this visualization is only used for illustration in the paper. The input to the convolutional neural network is not using a color-coded representation but uses the raw spectrogram matrix.

*RC: The discussion section is underdeveloped, lacks grounding and critique in the context of related literature and does not address the geomorphic significance of the*

*approach addressed.*

AC: We have addressed the fact that the discussion section is underdeveloped and expanded by adding two new subsections ("Feature Extraction" and "Overfitting") and extending the existing subsections. Moreover, we added more literature, such as (Walter et al., 2008), (Kuyuk et al., 2011), (Eibl et al., 2017), (Fei-Fei et al., 2006). Additionally, we have included more information in the evaluation section to support discussion about the possible advantages/disadvantages of our approach in regards to the geophysical application of event-based analysis. We show how our method can be used in our case study to extend event-based geophysical analysis. However, we find that a more detailed assessment and discussion about the results of our case study (in regards to geophysical significance) is out of scope of the submitted manuscript.

*RC: How does the constrained uncertainty of the approach considered compare to other sources of uncertainty, such as seismometer tilt (indeed, which component of the seismometer is being used, and why? – again see the work of Adam Young)*

AC: We have updated the manuscript to include which components of the seismometer are used (all three components, see Page 9, Lines 1-2). However, since we are not performing a device characterization study the analysis of tilt impact is not in the scope of our work.

*RC: [How does the constrained uncertainty of the approach considered compare to*

[Figure]

*other sources of uncertainty, such as] rock slope resonance and site effects (see e.g. Burjanek et al, 2012, 2017 GJI)? Some of the claims made about trade-off between time and accuracy feel poorly considered and require a more robust demonstration. There is also no discussion of the representativity of the case studies provided and how changes in the nature of the rock mass may affect the accuracy/source attribution of the seismic readings through time (e.g. resonance effects on duration and frequency as a rock mass degrades).*

AC: We accounted for changes in the nature of the rock by using a test set from a different year. We assume that the test set is representative for upcoming, never-seen-before data (see Discussion section "Overfitting" of the revised manuscript). Since the classifier shows good performance on the test set, the changes in nature of the rock are not assumed to have a significant impact on the performance of our classifier. This assessment is of course only valid in the scope of our dataset and an interesting future work would be to investigate how specific resonance or side effects have an impact on the accuracy of the classifier.

*RC: Specific comments (not exhaustive)*

AC: The following reviewer comments have been acknowledged and corrected but do not need a dedicated answer in our opinion.

*RC: There are many uses of e.g. in the manuscript – remove these and replace with 'such as' or 'for example' as appropriate.*
*RC: Brackets for citations are not always used correctly. Please check and amend.*

RC: Page 6 Line 28 - has monitored? Tense is not correct.
RC: [Page 7] Line 32 – sentence beginning 'Additionally' is not clear.
RC: Page 10 Line 5 – do not use comma splices (re: therefore)

AC: The passages the following comments refer to have been removed in the revised manuscript.

RC: [Page 2] Line 3 – HAS been demonstrated [Page 2] Line 5 – micro-seismic RECORDS?
RC: [Page 2] Line 6 – biased assessments of what? [Page 2] Lines 3 – 4 and 5 – 9 – very repetitive.
RC: [Page 2] Line 22 – its (not it's)
RC: [Page 2] Lines 18 – 19 – Do you mean the accurate attribution of seismic events?

AC: The following specific comments are replied to individually.

RC: Fig. 5 - what are the red/purple circles? Are these triggered microseismic events? This isn't clear in the figure or the caption.

AC: The red circles indicate the timestamps of the STA/LTA triggers we use for the paper. We have updated the caption accordingly.

RC: Page 1 Line 19 – check terminology. Rockfalls are a type of landslides (see e.g. Varnes, 1978, and subsequent iterations of this work).

AC: We acknowledge a loose usage of terminology and a possible sources of misunderstanding given our formulation and thus we have rewritten the passage mentioned. However, we would like to highlight that in Varnes, 1978 and in the subsequent work it is recommended that the term slope movement is used instead of landslides to avoid confusion. Consequently, you are right that rockfalls are a type of slope movement.

*RC: Page 2 Line 1 – what is the difference between acoustic emission and micro-seismic emission? Clarify.*

AC: The difference is the frequency range in which the emission is detected. The particular sentence related to this question was rewritten in the revised manuscript. Now, acoustic emission is not mentioned anymore to avoid confusion since the focus of the study is only on micro-seismic emission.

*RC: [Page 2] Line 9 – expand on scaling issues.*
*RC: [Page 7] Line 8 – expand on 'scaling issues' – this is unclear.*

AC: By scaling issues we mean that it is for example unfeasible to manually analyze and annotate continuous micro-seismic recordings of many years. We have reformulate a similar statement to "... manual methods suffer from their inability to scale to increasing data volumes ..."

*RC: [Page 2] Line 21 – what is the significance of footsteps?*

AC: The paragraph has been changed in the revised manuscript. It is now made clear that the STA/LTA event detectors can be used to register external influences, such as footsteps, but "cannot reliably discriminate geophysical seismic activity from external (unwanted) influence factors"

*RC: Page 3 Line 16 – F1 is not defined at this point. Indeed, much of the terminology in this section (e.g. ensemble classifier) is not clearly defined.*

AC: We have addressed this point by defining F1 before it is used in the text (except for the abstract where the value is required to make a statement about the performance of our method). Additionally, it is made clear than an ensemble classifier consists of multiple classifiers.

*RC: Page 7 Lines 6/7 – clarify 'sampling rate' – how was the sampling done? Or are you referring to the data transmission interval?*

AC: The sampling is done by performing a measurement every two minutes with the respective sensor and then transmitting that measurement via the wireless sensor

network to the server. We have adjusted the phrasing to make this aspect clear. To answer your question: In our case the sampling rate equals the inverse data transmission interval.

*RC: Lines 26 – 29 – the distinction between acoustic events and seismic events is confusing, seems a little arbitrary and lacks reference to the literature; some of the terms do not follow some conventions in e.g. laboratory monitoring of acoustic emissions; this is important for a contribution to the geophysical literature. These definitions and distinctions also come too late in the manuscript, since these terms are used earlier.*

AC: We acknowledge that given the geophysical background a consistent usage of terms is required. We have updated our definitions such that an event is defined as a trigger from a STA/LTA event detector. An event can have sources of geophysical or non-geophysical nature. In our study we apply a systematic method to identify non-geophysical sources in order to take them into account when analyzing geophysical sources. The definitions are now defined at the beginning of the paper and are consistently used throughout the manuscript

*RC: Table 2 – this needs a lot more detail – what is this showing?*

AC: The table caption has been updated to explain the table content, the structure of the neural network, its layers, strides and output channels.

*RC: [Page 6] Line 19 – triaxial or three-axis?*

AC: We use the word three-component in the revised manuscript

*RC: [Page 7] Line 11 – The microseismic records considered in the case study were affected?*

AC: It has been updated to "The recordings of the case study were affected..."

*RC: [Page 7] Line 24 – is 'sounds' the correct word here?*

AC: The passage the comment refers to has been removed in the revised manuscript.

*RC: Table 1 – Reword the caption. It is not the case that none of the other categories 'apply'. Rather, it is where you have not been able to classify the signal as one of the three categories discussed.*

AC: It has been updated to "when none of the other categories could have been identified".

*RC: Page 9 Line 9 – Figure 5(e) does not show an example of a rockfall. It shows an example of the seismic signature of a rockfall event.*

AC: It has been updated to "Shown are seismic signatures of ..."

---

## Author Comment (AC2) · 16 Nov 2018

*RC: Referee Comment*
AC: Author Comment

[Figure]

**1 Response to RC2**

AC: We thank the anonymous reviewer for the extensive review and the hint to additional literature. In the following we will address the comments and hint to the improvements in our manuscript. For a preliminary revised manuscript highlighting all the modifications please refer to the response to the editor.

*RC: This paper proposed a new method for identifying external influences such as winds or mountaineers in micro-seismic recordings. Because the external influences may cause bias interpretations, its identification is very important for understanding micro-seismic recordings. In addition, the method may help to interpret the external influences which are keys to improve our understanding of rock-slope failure processes. The similar idea using machine learning is already applied to seismic wave discrimination such as Li et al., 2018. This study is interesting and suitable for the publication after moderate revisions. I suggest the authors revise this manuscript and pay attention to the following list as general suggestions:*
*RC: 1. Acknowledge previous studies on this topic or related topics and make sure the readers understand your contribution;*

AC: We have acknowledged them. Moreover, we have rewritten the introduction and now the contributions are clearly demonstrated.

*RC: 2. Introduce more about methods especially for Convolution Neural Network since readers may not be familiar with this method at all;*

AC: We have added a new subsection "Convolutional Neural Networks" (starting Page 9, Line 7) which introduces convolutional neural networks and we have referenced additional literature to provide the reader with more background information on neural networks.

*RC: 3. Discuss more future works such as how to automatically learn signal pattern in the external influences to improve the classification and interpretation of rock-slope failure processes;*
*RC: Page 20 line 14-22: The method is trained based on negative examples. But in most conditions, we should pay more attention to the phenomena of interest. In the discussion part, the author should discuss how this method can improve our interpretation of the processes of interest.*

AC: We have added another paragraph to the discussion subsection "Classification of Negative Examples" (starting Page 23, Line 20) in which we explain how the method can improve the interpretation of the phenomena of interest. Moreover, we have extended our outlook (Page 26, Line 9) to include other methods which can improve the classification of external influences for example using semi-supervised and one-shot classification.

*RC: Here are more specific comments:*

*RC: Page 9, line 1-4: I am confused about this part. Does this part mean that the dataset may be mislabeled due to fog, lens flares or other reasons? Have this data been included in the training dataset? Similar problem for the rockfalls in line 8-10?*

AC: In case of limited visibility the images are not mislabeled, since the label represents what the annotator sees. However, since the seismic data is labeled with the help of images a certain probability of mislabeled samples exist if only images are used for annotation. In our case we reduce this probability by using an experienced annotator who can identify mountaineers on spectrograms and by using image sequences for annotation (before/after) when applicable. In the case of rockfalls we can only annotate time periods where we have additional information. Therefore it is most likely that we were unable to annotate all rockfall occurrences. As a consequence we did not consider a rockfall classifier. We have added this information on (see Page 9, Lines 6 - 9; Page 10, Lines 9 - 10).

*RC: Page 11, line 12-15, the dataset including training dataset and test dataset seems to be small and may have serious overfitting problem. The authors need to address this issue during the discussion part and prove the trained model can handle it well.*

AC: In the initial manuscript we have addressed the problem of overfitting in several paragraphs. In the revised manuscript we added another subsection "Overfitting" (starting Page 25, Line 31) to the discussion section explaining the impact of overfitting in our study.

[Figure]

*RC: Page 14, line 12-14 The results with ten iterations are presented in this paper, but it will be better to show how the results change for a different number of iterations (such as 1, 5, 10, 20 iterations).*

AC: We extended the evaluation section and have evaluated the impact of different training/test iterations in the "Classifier Evaluation" section (Page 20, Lines 9 - 19) and in Figure 11. We can confirm our choice of ten iterations.

*RC: Page 15, Line 10: The learning rate is very small, which may make the code very slow. Is there any specific reason to set this small value?*

AC: The value is the outcome of a preliminary hyper-parameter search and has been fixed for classifier training. Since the number of required iterations until convergence is rather small in comparison to other datasets/networks we found it reasonable to use such a small learning rate without negative impact regarding the total training duration. We have added the information explaining the learning rate to the revised manuscript (Page 16, Line 23 - 24).

*RC: Page 16, line 10-16: Since it needs to manually relabel for the dataset in some cases, it will be worth to discuss how the potential human errors during data labeling will influence the classifier performance.*

AC: We introduce a new evaluation which investigates how false labels affect the

classifier performance. This evaluation is presented and discussed in the section "Classifier Evaluation" (Page 20, Lines 20 - 22) and in Table 3.

---

## Author Comment (AC3) · 16 Nov 2018

*EC: Editor Comment*
AC: Author Comment

[Figure]

**1 Response to EC1**

AC: Dear Jens Turowski
thank you very much for your feedback. We addressed your comments in the following and responded to each referee individually. Attached to this response you find a preliminary version of the revised manuscript highlighting all modifications.

*EC: Dear authors,*
*I agree with the reviewers that the papers presents a potentially very useful contribution, however, currently there are weaknesses and the presentation (language and structure).*
*EC: When revising the paper, I ask you specifically to think from the perspective of a potential user. Is all the information available to reproduce the analysis? Can the reasoning be followed easily? Can the necessary information be clearly accessed in the manuscript? Is the approach described separately from specific features of the case study? Is the evaluation of the method objective?*

AC: We provided additional information by clarifying parameters, such as the impact of STA/LTA settings, learning rate, training iterations. We updated the structure of the manuscript, especially in the introduction and concept sections, to make it simpler to follow and access the presented information. To take especially the concerns of reviewer one into account we defined precisely what belongs to our method and what to the case study. In addition, we are more precise in the terminology we use and the geoscientific field we consider. In the evaluation we objectively demonstrate the benefits our method and detach evaluation of the case study from the evaluation of our method. We purposely leave out any geophysical interpretation of the results we

obtain after applying our method to the case study.

*EC: The comments of the reviewers largely speak for themselves; although they assess the paper from different angles, the common feature of their assessment is that there is a lack of clarity.*

AC: We addressed this issue by defining precise terminology at the beginning of the revised manuscript which is used persistently throughout the manuscript. Moreover we avoid misleading formulations. Moreover, we condensed significant parts of the introduction and reorganized the structure of the paper.

*EC: Reviewer #1 focuses more on the language and structure problems. In addition, he asks to evaluate your work in a broader context in the discussion/conclusion.*

AC: In addition to restructuring the first part of the manuscript, including updates to figures, we changed the discussion to include a broader . Specifically we introduced two new subsections ("Feature Extraction" and "Overfitting") and extended the existing subsections. Additionally, we added more literature to the discussion. Moreover, we gave hints to possible future works in the evaluation, discussion and outlook.

*EC: Similarly, reviewer #2 asks for a more detailed description of the method and a*

*better acknowledgment of previous work. I largely agree with that.*

AC: We added a more concise method description and introduced a new subsection "Convolutional neural networks" to explain the concepts of convolutional neural networks. Moreover, in addition to new geoscientific literature we acknowledge additional literature from the machine learning for seismology field.

*EC: In addition, I want to highlight a few specific stylistic points, which may help you to revise the paper and achieve the aims outlined above. I will do this quoting specific example (page.line). 2.11 ' ... very unattractive overall solution ... ': please avoid subjective judgements such at this. It is better to list the pros and cons, and then explain your priorities and your reasoning. 5.3 'Care is taken ... '; 6.12 'But the meticulous care does pay off.'; 7.8 'Care has been taken to prevent significant data gaps ... ': Such statements are not helpful to the reader. The phrasing is meant to convey some particularly high standards of scientific rigour. However, it is unclear what you have actually done, and thus your 'care' is not reproducible. It would be better if you describe your actual measures (e.g., for preventing significant data gaps) and then describe how well they worked, and if they failed, why they failed. Again, here it is important to keep the reproducability of the work in mind.*

AC: We have worked through the paper to remove any subjective judgment, imprecise wording or not well-documented phrases. For example we have updated the example you mention to "Significant data gaps are prevented by using solar panels, durable batteries and field-tested sensors..."
*EC: 8.6 'It becomes apparent in Fig. 5 (b)-(c) that anthropogenic noise, such as mountaineers walking by or helicopters, can have a strong influence on seismic recordings.': This sentence is an example of how results are mixed into the method description. There are multiple other instances. I ask you to separate this and present the methodology in the methods section and the results in the results section.*

AC: We corrected your example and reformulated the sentence. In addition, we have restructured certain parts of the manuscript to avoid a mix-up of methods and results. For example we have placed all of our findings of the statistical evaluation into the results section.

*EC: 18.2 'The results of the classifier experiments from Sect. 3.2 are listed in Table 3.': Such sentences contain little information. It would be better to state the main result or feature (that is important in the current context) and then cite the table in parentheses. Looking forward to seeing your revised paper, best wishes, Jens Turowski*

AC: We have reformulated the presented example and have worked through the paper to address similar issues. For example we updated the example you mention to "The results of the classifier experiments (Table 2) show that ..."

---

## Referee Report (RR1)

The authors have addressed all my comments for this paper and answered the technical questions I have for this method. The paper has been significantly improved after revising. But I still have two minor suggestions for the authors about figures. I will recommend the editor accept this paper and publish it after they are addressed.

(1)    I know this is a technology/methodology paper, but I suggest the authors add a subplot in Figure 3 to show the location of the study site in a map view. In that case, the readers know where the study site is.
(2)    I understand these red dots in Figure 5 indicate the timestamps of triggers. However, I found it's very difficult to tell in this Figure, especially for subplot a and b. I think using arrows or another way will be better than this.

---

## Author Response (AR2)

**Letter to the editor**

Dear Editors,

We thank for the constructive and positive feedback. The referees state the study as very interesting and further highlight the research as significant. However, they also identified a few minor issues which mainly concern minor language/editorial changes and some clarification. In the revised manuscript we addressed all the referees comments and added in the general response one by one explanation to the points raised by the referees.

With kind regards,
Matthias Meyer
On behalf of all authors

*RC: Referee Comment*
AC: Author Comment

**Response to Anonymous Referee #2**

AC: We thank Anonymous Referee #2 for the review and suggestions for improvement. The response by the authors to the reviewer comments are listed and explained.

*RC: I know this is a technology/methodology paper, but I suggest the authors add a subplot in Figure 3 to show the location of the study site in a map view. In that case, the readers know where the study site is.*

AC: We addressed this comment in the revised manuscript (Figure 3).

*RC: I understand these red dots in Figure 5 indicate the timestamps of triggers. However, I found it's very difficult to tell in this Figure, especially for subplot a and b. I think using arrows or another way will be better than this.*

AC: We acknowledge the suggestion to replace the red dots (which represent triggered events) in Figure 5 by arrows. However, as there are so many triggered events, the visualization with arrows is worse. Therefore, we decided to stay with the red dots.

*RC: Referee Comment*
AC: Author Comment

**Response to Referee#3: Marine Denolle**

AC: We would like to thank Marine Denolle for the extensive review and the valuable feedback. We will incorporate the feedback and address all comments in the following. For a preliminary revised manuscript highlighting all the modifications please refer to the response to the editor.

*RC: I read that the seismic data is multi component. But how was it included in the MicroseismicCNNs, in a 3-channel input like in Perol et al, 2018?*

AC: Yes, the input has 3-channels. For each component (channel) the time-frequency representation is calculated and the resulting 2D matrix represents one of the three input channels. We added this information to the revised manuscript (P15L2f):
*"The proposed convolutional neural network (CNN) to identify non-geophysical sources in micro-seismic signals uses a time-frequency signal representation as input and consists of 2D convolutional layers.* **Each component of** *the time-domain signal, sampled at 1 kHz, is first offset-compensated and then transformed with a Short-Time Fourier Transformation (STFT). Subsequently, the STFT output is further processed by selecting the frequency range from 2 to 250 Hz and subdividing it into 64 linearly-spaced bands.* **This time-frequency representation of the three seismometer components can be interpreted as 2D signal with three channels, which is the networks input.***".

*RC: P1L4: "wide range of applications", can you list them instead of just citing?*

AC: We addressed this comment in the revised manuscript (P1, L1):
*Passive monitoring of elastic waves, generated by the rapid release of energy within a material (Hardy, 2003) is a non-destructive analysis technique* **allowing a wide range of applications in material sciences (Labuz et al., 2001), engineering (Grosse, 2008) and natural hazard mitigation (Michlmayr et al., 2012) with recently increasing interest into investigations of various processes in rock slopes (Amitrano et al., 2010; Occhiena et al., 2012).**

*RC: P2L30: citation for STA/LTA is Allen 1978, as you note later. It's worth citing it now.*

AC: We addressed this comment in the revised manuscript.

*RC: P3L5–18: In general, in the seismology community, some of these discussions have been introduced as the difference between unsupervised learning techniques (auto-correlations, FAST Yoon et al, 2018; PageRank Aguiar and Beroza 2014, sta/lta Allen 1978), and supervised learning techniques (template matching using cross correlation as a distance metric for instance, and other ML classifier).*

AC: We rephrased the paragraph based on your suggestions and included the suggested literature (P3L8–11):
*"There exist several algorithmic approaches to mitigate the problem of external influences by increasing the selectivity ofevent detection.* **Unsupervised algorithms such as auto-correlation (Brown et al., 2008; C. Aguiar and C. Beroza, 2014; Yoon et al., 2015), but these are either computationally complex or do not perform well for low signal to noise ratios. Supervised methods can find events in signals with low signal to noise ratio. For example template matching approaches such as** *cross-correlation methods (Gibbons and Ringdal, 2006)..."*

*RC: P3L8–9–10: I am not sure why the sentence on ML technique is separated from neural network techniques. As you are pointing out, the ground truth data, namely the quality of the labeled data in supervised learning, is tricky. It's okay with large earthquake catalogs (like in the Ross et et al 2018; Kong et al, 2016), but it remains indeed difficult for non-labeled environmental or anthropogenic factor.*

AC: We updated the sentences to make clear that neural networks are in fact a subcategory of machine learning. We also incorporated the additional literature you provide (P3L12–16):
*"The most recent* **supervised** *methods are based on machine learning techniques (Reynen and Audet, 2017; Olivier et al., 2018)* **including the** *use of neuralnetworks (Kislov and Gravirov, 2017; Perol et al., 2018; Li et al., 2018; Ross et al., 2018).* **These learning approaches** *show promising results with the drawback that large datasets containing ground truth (verified events) are required to train these automated classifiers. In earthquake research large databases of known*

*events exist* **(Kong et al., 2016; Ross et al.)**, *but ..."*

*RC: Figure 1: I don't think it's necessary. Figure 2 contains some of this content.*

AC: We appreciate this comment. Nevertheless, we would prefer to keep this figure in the manuscript as it is a simplification of the analysis method and thereby supports readers who are new in this subject.

*RC: P5L15: "used" is utilized twice in the same sentence*

AC: We addressed this comment in the revised manuscript: *"The seismometer* **applied** *in the case study presented is used to assess the seismic activity by ..."*

*RC: P14L7: "the the" (removed one of them)*

AC: Done.

*RC: P15L9: "Chollet and others" -¿ et al.*

AC: Done.

*RC: P17L15–16: why dropping the other classes and why not using a binary classifier?*

AC: Training including the other classes resulted in better performance. For example without these additional information the classifier was bad at distinguishing lens flares from mountaineers and training a classifier was impossible. We addressed this comment in the revised manuscript by rephrasing the paragraph (P17L20–24):
*"The network has been trained to detect 5 different categories.* **In this paper only the metrics for mountaineers are of interest for the evaluation and the metrics for the other labels are discarded in the following. However, all categories** *are relevant for a* **successful training** *of the mountaineer classifier. These categories consist of mountaineer, low visibility (if the lens is partially obscured), lens flare, snowy (if the seismometer is covered in*

*snow) and bad weather (as far as it can be deduced from the image)."*

*RC: P20L16: 1.5 x IQR, use a different font for "times" so that it does not look like the "x"*

AC: Done.

*RC: P20L17: "of of", remove one "of"*

AC: Done.

[revised manuscript text omitted]